# High-order thalamic inputs to primary somatosensory cortex are stronger and longer lasting than cortical inputs

Wanying Zhang[1], Randy M Bruno[1,2,3]*

[1]Department of Neuroscience, Columbia University, New York, United States; [2]Kavli Institute for Brain Science, Columbia University, New York, United States; [3]Zuckerman Mind Brain Behavior Institute, Columbia University, New York, United States

**Abstract** Layer (L) 2/3 pyramidal neurons in the primary somatosensory cortex (S1) are sparsely active, spontaneously and during sensory stimulation. Long-range inputs from higher areas may gate L2/3 activity. We investigated their in vivo impact by expressing channelrhodopsin in three main sources of feedback to rat S1: primary motor cortex, secondary somatosensory cortex, and secondary somatosensory thalamic nucleus (the posterior medial nucleus, POm). Inputs from cortical areas were relatively weak. POm, however, more robustly depolarized L2/3 cells and, when paired with peripheral stimulation, evoked action potentials. POm triggered not only a stronger fast-onset depolarization but also a delayed all-or-none persistent depolarization, lasting up to 1 s and exhibiting alpha/beta-range oscillations. Inactivating POm somata abolished persistent but not initial depolarization, indicating a recurrent circuit mechanism. We conclude that secondary thalamus can enhance L2/3 responsiveness over long periods. Such timescales could provide a potential modality-specific substrate for attention, working memory, and plasticity.
DOI: https://doi.org/10.7554/eLife.44158.001

*For correspondence:
randybruno@columbia.edu

**Competing interests:** The authors declare that no competing interests exist.

## Introduction

Layer (L) 2/3 pyramidal neurons in the primary sensory cortices exhibit sparse activity, both spontaneously and in response to sensory stimuli (*Barth and Poulet, 2012*). Even in awake animals performing simple tactile detection tasks, L2/3 firing probability remains low (*O'Connor et al., 2010*). In the whisker representation (barrel cortex) of rodent primary somatosensory cortex (S1), sensory inputs during active whisking reliably evoke short-latency subthreshold depolarization in L2/3 pyramidal neurons (*Crochet et al., 2011*; *Sachidhanandam et al., 2013*). Sensory input also engages strong feed-forward inhibition, which keeps membrane potential ($V_m$) of most L2/3 neurons below spike threshold, rendering them quiet or only sparsely responsive (*Crochet et al., 2011*). Low firing rates in L2/3 are unlikely to be due to highly selective receptive fields: We recently showed that presenting complex spatio-temporal patterns of whisker stimulation optimized for individual neurons strongly engages neurons in L4-6, but not L2/3 (*Ramirez et al., 2014*). These previous studies suggest that ascending sensory information arriving in L2/3 from L4 alone may be insufficient to drive L2/3 activity.

Excitatory inputs from other brain regions, perhaps activated under specific behavioral conditions, may be required to engage L2/3. L2/3 neurons in S1 receive inputs from higher-order cortical and subcortical regions, such as the primary motor cortex (M1) (*Kinnischtzke et al., 2014*; *Lee et al., 2013*; *Petreanu et al., 2009*; *Veinante and Deschênes, 2003*) and the secondary somatosensory nucleus of the thalamus, called the posterior medial (POm) nucleus (*Jouhanneau et al., 2014*; *Lu and Lin, 1993*; *Ohno et al., 2012*; *Rubio-Garrido et al., 2009*;

*Sherman and Guillery, 2011*; *Wimmer et al., 2010*). Previous functional studies have mainly characterized the synapses from M1 or POm to S1 in vitro (*Kinnischtzke et al., 2014*; *Lee et al., 2013*; *Petreanu et al., 2009*). POm is known to be a potent driver of activity in secondary somatosensory (S2) in vitro (*Theyel et al., 2010*), and a recent in vivo study of anesthetized mice found that POm input could enhance the responsiveness of L5 pyramidal neurons to sensory stimulation (*Mease et al., 2016*). The response of the S1 L2/3 network to long-range synapses, particularly in vivo, has received little attention. Additionally S1 receives significant anatomical input from S2 (*Cauller et al., 1998*), whose effects remain largely unexplored.

We hypothesized that L2/3 neurons, though typically silent, might respond to sensory stimuli in conjunction with inputs from higher-order brain regions. Such a circuit could contextually modulate sensory responses. To compare the potential impact of M1, POm, and S2 inputs on sensory processing, we combined optogenetic stimulation of their synaptic terminals with in vivo whole-cell recording in S1. We discovered that POm activation elicited significantly stronger depolarizations in L2/3 neurons than M1 or S2 activation. Only POm input was able to boost sensory responses of L2/3 neurons in both anesthetized and lightly sedated animals. Furthermore, we discovered that POm activation in awake and sedated animals elicited long-lasting depolarization in L2/3 in an all-or-none matter. These results demonstrate a potential circuit mechanism by which POm can enhance L2/3 processing during behavior for prolonged periods.

## Results

### Laminar distributions of M1, S2, and POm axons

To compare long-range M1, S2, and POm inputs to rat barrel cortex, we injected an adeno-associated virus expressing a fusion of channelrhodopsin (ChR2) and yellow fluorescent protein (YFP) into each of these three areas. Three to four weeks post-injection, there was intense ChR2-YFP expression in the infected areas (*Figure 1A*, bottom). In all cases, labeled axons were observed in barrel cortex, with significant innervation of L1. Outside of L1, axonal distributions across varied based on the area of origin (*Figure 1A*, top): POm axons were concentrated in L4 septum and L5A; M1 axons resided mostly in deep L5B and L6; and S2 axons formed bands in both L5 and L6 (*Figure 1B*). Axons from all three regions avoided L4 barrels and were rare in L3. No labeled somata were ever

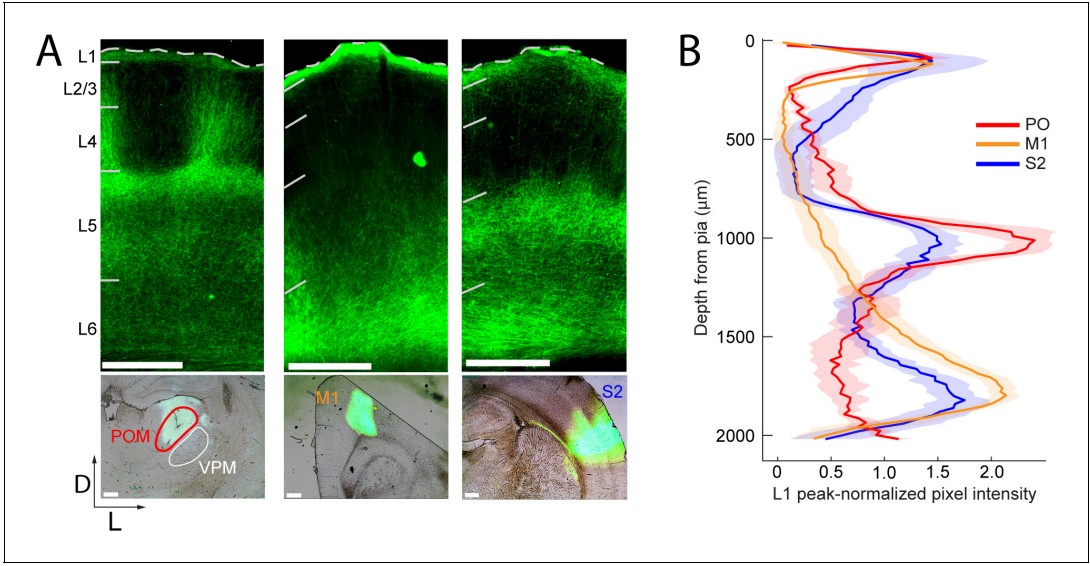

**Figure 1.** Long-range projection axons from POm, M1 and S2 innervate Layer 1. (A) Top panel: labeling of long-range projection axons with ChR2-eYFP from POm (left), M1 (center), and S2 (right) shown in coronal sections of barrel cortex. Bottom: Sites of viral infection in POm (left), M1 (center), and S2 (right). Bar: 500 µm. D, dorsal; L, lateral. Dashes, pia. (B) Average laminar profile of ChR2-eYFP labeled axons in barrel cortex. Pixel intensity was measured only for barrel-related columns (not including septal regions) and normalized to peak (n = 4 rats for each infected region; shading,±SEM).
DOI: https://doi.org/10.7554/eLife.44158.002

observed in barrel cortex, even after immunohistochemical amplification, ruling out retrograde infection of S1 neurons.

## L2/3 is more strongly depolarized by POm than M1 or S2

To assess the relative efficacies of these long-range inputs to barrel cortex, we initially recorded whole-cell from L2/3 barrel cortex pyramidal neurons in anesthetized rats. We measured the postsynaptic responses evoked by photo-activation of ChR2-containing axons (*Figure 2A*). Photo-activation (a single 10 ms pulse) of long-range fibers near recorded neurons elicited either fast-onset excitatory postsynaptic potentials (EPSPs) or no discernable responses (example cells in *Figure 2B*). Photo-activation of POm axons elicited EPSPs in the majority of recorded neurons (21 out of 33 neurons; *Figure 2C*), which were substantial in size (mean ± SEM, 4.2 ± 0.8 mV). By comparison, M1 and S2 inputs produced significantly smaller responses in L2/3 pyramidal neurons (*Figure 2C*) (M1: 5 out of 19 neurons, 1.2 ± 0.48 mV; S2: 15 out of 22 neurons, 1.3 ± 0.28 mV; one-way ANOVA, $p < 10-5= 0.006$). Photo-activation of POm axons was also significantly larger when the full population of neurons, with and without EPSPs, were analyzed (POm: n = 33, mean ± SEM 2.7 ± 0.6 mV; M1: n = 19, mean 0.16 mV ± 0.22 mV; S2: n = 22, mean 0.87 ± 0.23 mV; one-way ANOVA, p = 0.0015; *Figure 2C, D*).

Neurons recorded in POm-infected animals had baseline membrane potential ($V_m$; *Figure 2E*) and input resistance ($R_{in}$; *Figure 2F*) similar to those in M1- and S2-infected animals, indicating that the stronger POm input was not due to random variation in intrinsic properties among the three sampled groups. These $V_m$ and $R_{in}$ values are typical of L2/3 neurons in other in vivo whole-cell recording studies under similar conditions (*Brecht et al., 2003*; *Constantinople and Bruno, 2011*). Additionally, the kinetics of photo-activated EPSPs (onset latencies, rise times, and decay times) were similar across M1, S2, and POm experiments (*Figure 2—figure supplement 1*). Virus occasionally spread along the injection pipette track and into high-order visual thalamic nuclei LP and LD, superficial to POm (*Figure 1A*, bottom). These two nuclei, however, do not target barrel cortex (*Kamishina et al., 2009*; *van Groen and Wyss, 1992*) and are unlikely to explain our observations. Moreover, for animals in which multiple cortical neurons were patched, L2/3 cells with the strongest POm inputs could be observed in the same animals having cells receiving relatively weak POm input—up to 10 times smaller (*Figure 2—figure supplement 2*). This indicates that the strongest POm inputs are unlikely to result solely from variability in injection placement or viral efficiency and, conceivably, may reflect L2/3 cell variation.

## POm facilitates sensory responses of L2/3 pyramidal neurons

The greater potency of POm input relative to S2 and M1 input suggests a unique role in sensory processing. To compare the impact of each long-range input on sensory processing, we recorded L2/3 neuron responses to principal whisker (PW) stimulation, photo-activation of axons, or the combination of the two (*Figure 3A*). Delivery of the whisker stimulus preceded the light by 7 ms to account for known synaptic delay from the periphery to L2/3 and thereby maximize the coincidence of elicited potentials. Comparison of the combined stimuli and whisker stimulus alone showed that photo-activation of long-range axons from any of the three regions rarely suppressed sensory responses if at all (*Figure 3B,D*). Only POm projections (*Figure 3B*, red) ever facilitated L2/3 neuron input. Neurons without discernable excitatory responses to light alone (occurring in all three groups) showed no facilitation of sensory responses by photo-activation (gray circles) and are not analyzed further. In most cells, combined light and sensory stimuli elicited sub-threshold responses that were similar to the linear sum of the cell's responses to each individual stimulus, suggesting that sensory input and each of these long-range inputs are linearly integrated passively by the L2/3 neurons (*Figure 3C*). This linearity also means that the enhancement of sensory response by light-induced input is directly proportional to the response to the light alone. An exception is the group of cells receiving the strongest POm inputs, which often exhibited sublinear integration (filled symbols) as expected from basic cellular properties (i.e., reduced driving force). L2/3 neurons in anesthetized animals lacked clear spiking responses to whisker stimuli as observed previously (*Brecht et al., 2003*). Nevertheless, these subthreshold results raised the possibility that POm may be an important controller of L2/3 activity under other conditions.

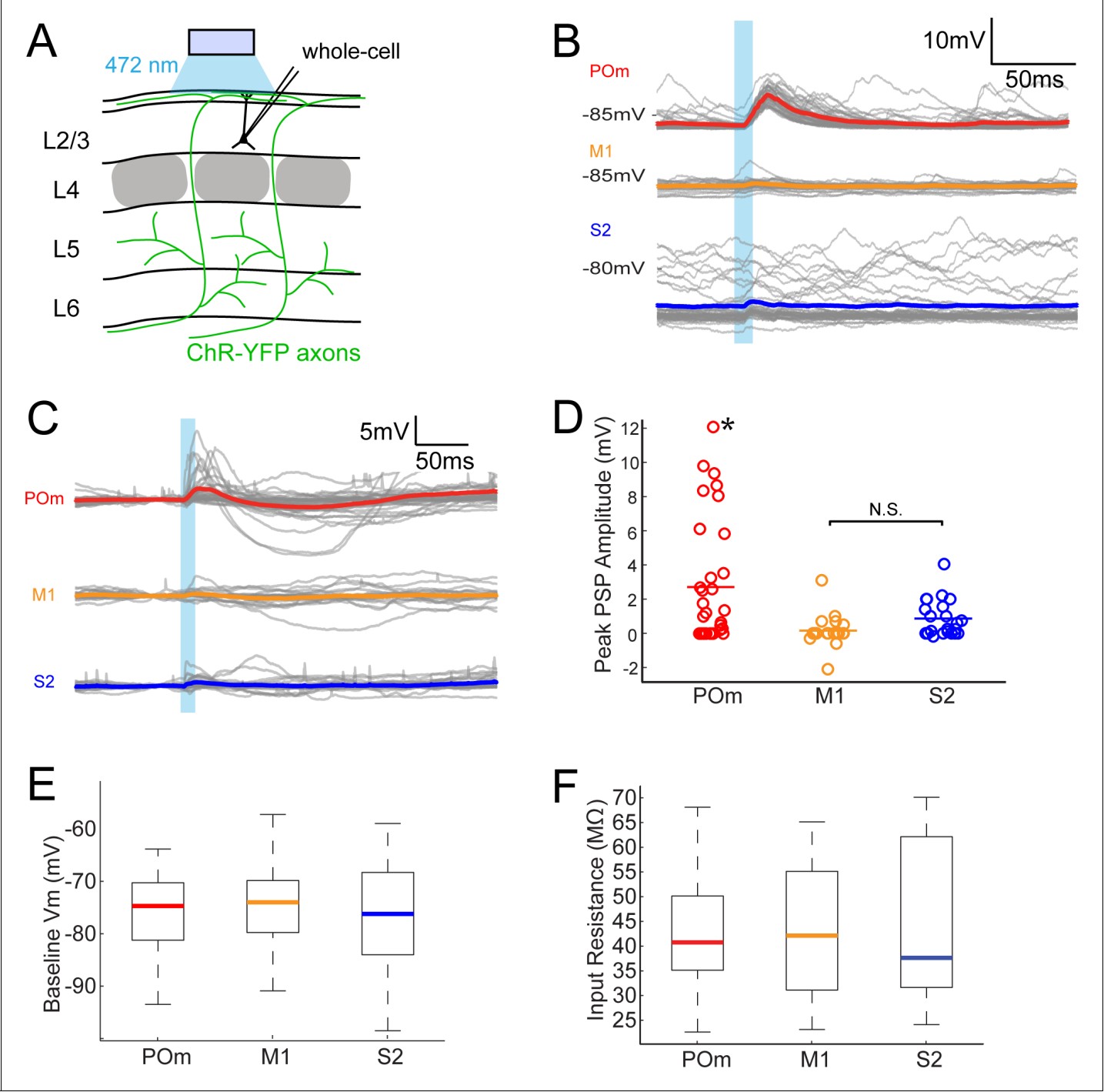

**Figure 2.** POm axons more strongly excite L2/3 pyramidal neurons than M1 or S2 axons do. (A) Schematic of in vivo ChR2 activation of long-range projection axons in barrel cortex and whole-cell recording of L2/3 pyramidal neurons. Gray, L4 barrels. (B) Example whole-cell responses of three L2/3 pyramidal neurons to photo-activating long-range axons from infected cells in POm (top), M1 (middle), and S2 (bottom). Gray, 20 single trials recorded per cell; Colored, trial average. (C) Population average, baseline-subtracted responses of L2/3 pyramidal neurons to photo-activation of axons. Gray, average responses of individual neurons, n = 33 for POm, 19 for M1, and 22 for S2. Colored, average response for each input type. Baselines just prior to photo-activation have been aligned to allow visual comparison of responses. (D) Distributions of peak response amplitudes to photo-activation. Activation of POm axons on average elicits a significantly larger excitatory response than activation of M1 and S2 . One-way ANOVA, p=0.0015. N.S., not significant. (E) Distribution of baseline $V_m$ of recorded L2/3 neurons. Baseline $V_m$ did not differ among groups (one-way ANOVA, p=0.92). (F) Distribution of resting $R_{in}$ of recorded L2/3 neurons. Resting Rin did not differ among groups (one-way ANOVA, p=0.85).

DOI: https://doi.org/10.7554/eLife.44158.003

*Figure 2 continued on next page*

*Figure 2 continued*

The following figure supplements are available for figure 2:

**Figure supplement 1.** Kinetic characteristics of initial EPSP elicited by photo-activation.

DOI: https://doi.org/10.7554/eLife.44158.004

**Figure supplement 2.** Individual animals exhibited wide ranges of POm input strengths.

DOI: https://doi.org/10.7554/eLife.44158.005

State is well known to impact the firing patterns of neurons in the central nervous system. Under anesthesia, POm neurons fire at lower rates than during wakefulness (*Masri et al., 2008*). Anesthetics may also directly alter synaptic properties. We therefore repeated the experiment using an alternative preparation in which local anesthetics are combined with administration of a frontally-acting opiate (fentanyl) to relax the animal. Previous studies have shown that cortical dynamics, intracellular membrane potential, and sensory responses recorded in this sedated preparation resemble those recorded in awake animals (*Constantinople and Bruno, 2011*; *Simons et al., 1992*). POm axons continued to induce substantial EPSPs in sedated animals, even larger than that found in urethane-anesthetized animals (*Figure 4A*, red). M1 axons were also significantly more effective under sedation (yellow), but did not reach the same potency as POm inputs. S2 inputs were similarly weak under both conditions. Despite the larger magnitude of POm input, the pathway appears to be engaging significant inhibition: Regression of initial POm input magnitude against baseline membrane potential predicts a reversal potential of −34 mV, suggestive of mixed excitation and inhibition and consistent with other inputs to L2/3 pyramids in vivo (*Petersen et al., 2003*). In summary, POm input appeared overall more potent relative to S2 and M1 inputs under very different states— both light sedation and general anesthesia.

POm is directly adjacent to VPM, the primary thalamic nucleus for this system. The spread of infection into VPM is a potential confound. We measured the area of the POm infection in the section where the infection was largest, which is approximately the infection center. POm infections were an order of magnitude larger in cross-sectional area than any potential VPM contamination. PSP amplitude correlated with the size of the infected region in POm, not the one in VPM (*Figure 4—figure supplement 1*). Additionally, we observed no statistically significant relationship of POm input magnitude and recording depth from the pia (regression, p=0.18), indicating that L2 and L3 pyramids were similarly affected by photo-activation even though L3 pyramids are more likely than L2 pyramids to receive VPM input. Thus, intentional POm infection rather than unintended VPM contamination appears to account for the bulk of the synaptic input we measured.

Another possible explanation for the relative potency of POm input is the presence of POm axons in the septum between barrels (*Figure 1A*, left). If septal branches explained the stronger POm input, L2/3 pyramidal neurons closer to septa would be expected to receive stronger POm inputs because L4 axons tend to arborize directly superficial to their somata. We tested this possibility by measuring the horizontal distance of L2/3 somata from the center of the nearest barrel column. There was, however, no relationship of distance to the magnitude of POm input (*Figure 4B*; $R^2 = 0.06$, p=0.4), suggesting that septal innervation is not a major contributor to POm's relative potency. This result is consistent with anatomy: all individual POm axons reconstructed to date branch extensively in L1 but have few or no branches in the septum that they traverse to reach L1 (*Deschênes et al., 1998*; *Ohno et al., 2012*). Additionally, a single POm axon often ascends the septal spaces between more than one pair of barrels. Thus, a large number of unbranched axons are concentrated in the small septal volumes, potentially explaining why PSP size does not reflect high septal epifluorescence.

We did not detect any statistically significant enhancement of L2/3 neurons' responses to whisker stimulation by S2 or M1 axon stimulation (*Figure 4C,D*). Photo-activation of POm inputs appeared to slightly boost the sub-threshold responses of L2/3 pyramidal neurons to PW stimulation under sedation (*Figure 4C*) as under general anesthesia, but reached only trend-level significance. However, this slight subthreshold increase translated into substantial increases of suprathreshold (spiking) responses of those L2/3 neurons to sensory stimuli (*Figure 4D,E*). Spiking responses were increased six fold (mean ±SEM, whisker stimulus alone: 0.16 ± 0.09 Hz; whisker +light: 1.1 ± 0.36 Hz). Such large enhancements in spiking are expected from small increases in subthreshold input due to the

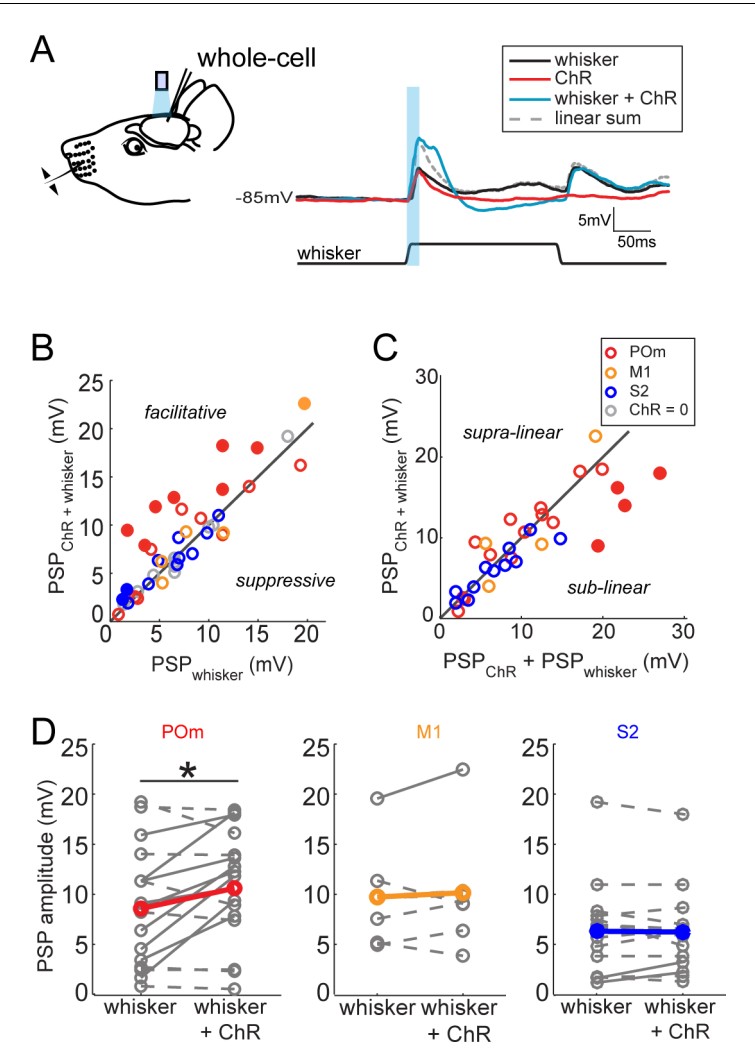

**Figure 3.** Activation of POm inputs facilitates sensory responses of L2/3 pyramidal neurons. (**A**) Left, schematic of in vivo ChR2 activation of long-range projection axons and whisker stimulus during whole-cell recording from a L2/3 pyramidal neuron. Right, average response of an example neuron to photo-activation alone (red), deflection of the PW (black), and simultaneous photo-activation of POm axons and PW deflection (blue). Dotted line, linear sum of PSP response to PW deflection and photo-activation. Light blue bar: 10 ms laser pulse. (**B**) Photo-activation of POm inputs to L2/3 facilitates the neurons' sensory responses. Filled circles, neurons for which the responses to the combined sensory and photo stimuli are statistically significantly greater than the sensory response alone. (**C**) L2/3 responses to simultaneous sensory and photo stimuli are linear. Line, linearity. Filled circles, neurons for which the responses to the combined sensory and photo stimuli are statistically significantly smaller than the sum of responses to sensory or photo stimulation alone. (**D**) Only photo-activation of POm axons significantly facilitates sensory responses of L2/3 pyramidal neurons (paired t-test: POm, n = 16, p=0.027; M1, n = 5, p=0.86; S2, n = 13, p=0.95). Dashed lines, cells for which whisker response and combined whisker and photo response are not statistically different. Solid lines, cell for which responses are statistically different.
DOI: https://doi.org/10.7554/eLife.44158.006

nonlinear nature of spike generation. Sublinear integration of POm input that remains net facilitatory mirrors the sublinear, net facilitatory effects we have previously shown for sensory surround stimulation (*Ramirez et al., 2014*).

We conclude that, of several long-range pathways innervating primary somatosensory cortex, POm input is relatively more potent to L2/3 pyramidal neurons than S2 or M1 inputs. Our results suggest that POm may be gating L2/3 suprathreshold sensory responses.

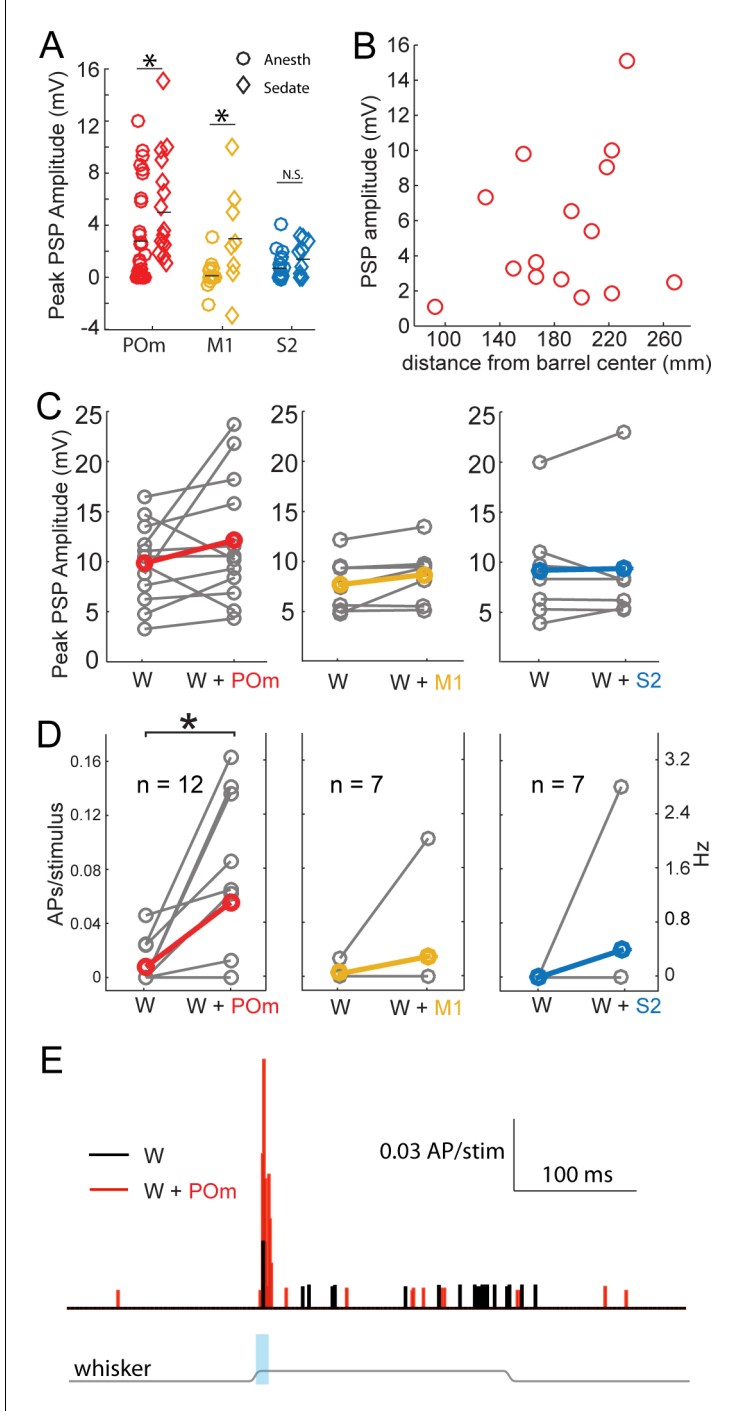

**Figure 4.** Strong excitation of L2/3 neurons by photo-activation of POm axons is not an artifact of general anesthesia. (A) Average L2/3 responses to photo-activation of POm and M1 axons under fentanyl sedation is larger than those measured under urethane general anesthesia (two-sided rank sum tests; POm, n = 15 fentanyl cells, mean ± SEM 5.5 ± 1.0 mV, p=0.004; M1,n = 8 cells, mean 3.1 ± 1.4, p=0.009; S2, n = 9 cells, mean 1.6 ± 0.4, p=0.13). Lines, means. (B) PSP amplitude (in response to POm activation) as a function of distance of L2/3 soma from center of nearest barrel column. (C) Under fentanyl sedation, neither POm (left) nor M1 axon activation (middle) significantly boosted subthreshold whisker responses (paired t-tests; n = 12, p=0.16 and n = 7, p=0.07 respectively) despite slight appearances of trends. S2 axon activation (right) did not significantly boost the subthreshold whisker responses (n = 7, p=0.83). Gray, individual cells; Red, mean. (D) Left: Under fentanyl sedation, POm axon activation significantly increase spiking responses of L2/3 neurons to whisker deflection

*Figure 4 continued on next page*

*Figure 4 continued*

(paired t-test, n = 12, p=0.018). Middle: M1 axon activation does not significantly increase spiking responses of L2/3 neurons to whisker deflection (n = 7, p=0.36). Right: S2 axon activation does not significantly increase spiking responses of L2/3 neurons to whisker deflection (n = 7, p=0.37). Gray, individual cells; Red, mean. Some data points have the same values (i.e., 0). (E) Population PSTH of spiking responses of L2/3 neurons to whisker deflect (black) and whisker deflection combined with POm axon activation (red), n = 12 neurons. Gray line: displacement of whisker stimulus. Blue bar: duration of photostimulation (when present).

DOI: https://doi.org/10.7554/eLife.44158.007

The following figure supplement is available for figure 4:

**Figure supplement 1.** Strength of ChR-mediated input to L2/3 is related to POm not VPM infection.

DOI: https://doi.org/10.7554/eLife.44158.008

## Persistent depolarization under sedation and wakefulness

The time course of L2/3 responses during POm activation is also unique. In anesthetized animals, PSPs induced by photoactivation of POm inputs return to baseline 50–100 ms after the cessation of light stimuli (*Figures 2* and *3*) as recently observed (*Gambino et al., 2014*). In sedated animals, we discovered a persistent depolarization in L2/3 pyramidal cells lasting an order of magnitude longer (*Figure 5A*). After the initial fast-onset EPSP and a period of pronounced inhibition, persistent depolarization began ~150 ms after POm axon activation (mean onset, 163 ± 13 ms) and typically continued for up to one second (mean duration, 758 ± 133 ms; *Figure 5B*, red). For each neuron recorded, ~80% of trials showed a prolonged response, which was unseen in anesthetized animals. M1 and S2 inputs did not elicit persistent depolarization (yellow and blue).

To check whether the different time courses of POm, M1, and S2 were a trivial consequence of POm's stronger input, we recorded from animals in which we infected the primary thalamic nucleus (VPM). VPM is an even more powerful driver of S1, including L4 excitatory neurons, which are themselves highly active and mainly target L2/3 basal dendrites (*Bruno and Sakmann, 2006*; *Egger et al., 2008*; *Lübke et al., 2003*; *Viaene et al., 2011a*). Strikingly, axons from VPM elicited little or no persistent depolarization in L2/3 (green), despite generating large initial responses (13.4 ± 3.2 mV; n = 9) that were on average 7-fold larger than those elicited by POm. Thus, persistent depolarization of cortex appears to be a unique capability of secondary thalamus, rather than being a consequence of the magnitude of input generated by a pathway.

On individual trials, the persistent depolarization showed clear periodicity in the 10–20 Hz frequency band (*Figure 5A,C*, red). Signals in the 10–20 Hz range were absent from corresponding times during VPM, M1 and S2 axonal photo-activation (green, yellow and blue) as well as spontaneous activity (black). Thus, 'alpha' (8–15 Hz) and 'beta' range (12.5–30 Hz) oscillations in cortex appeared to be unique to stimulation of POm axons.

The persistent depolarization is also detectable in L2/3 recordings of local field potentials (LFPs), which have inverted polarity relative to whole-cell recordings (compare *Figure 6A,B*). We exploited the LFP to explore how other conditions might influence responses to POm. In our initial experiments (*Figures 2–5*), we had used high light intensity to ensure that the weak responses of M1 and S2 were real. Varying light intensity over 0.1–35 mW, we found that the persistent depolarization elicited by POm axons appeared at all intensities that evoked an initial response (*Figure 6D*). Whereas the initial peak scaled with light intensity (*Figure 6E*, top), the persistent depolarization was virtually all-or-none (middle), appearing with as little as 0.6 mW. The all-or-none quality of the depolarization raises the possibility that POm may not exist to provide strong, precise input to L2/3, but rather to enhance L2/3 activity over long time windows.

To verify that the persistent depolarization produced by this pathway is relevant for awake animals, we recorded LFPs in L2/3 of awake head-fixed rats. LFP responses to POm in awake rats showed the same characteristic initial and persistent depolarization (*Figure 6C*) seen in sedated animals (*Figure 6B*). We conclude that during wakefulness POm is able to open an unusually long window of L2/3 depolarization.

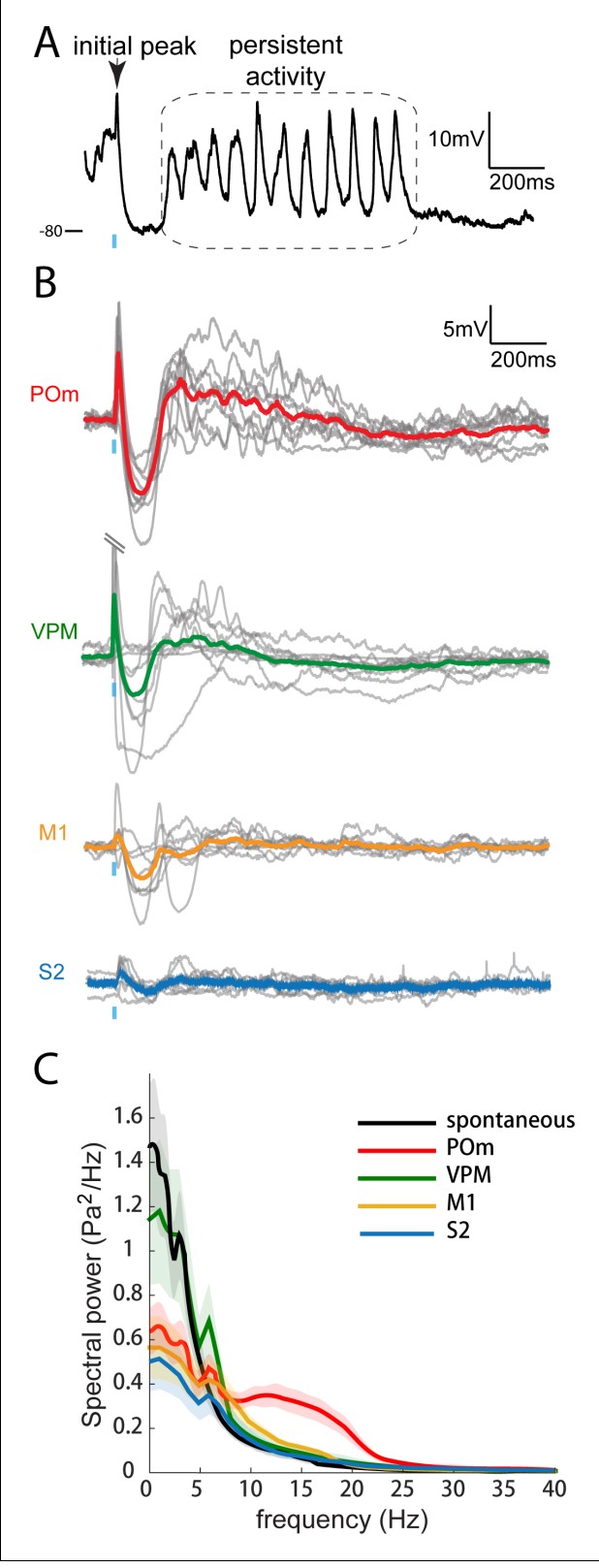

**Figure 5.** Only POm inputs elicit large late, persistent depolarization in L2/3 neurons under fentanyl sedation. (**A**) Example recording (single trial) of a L2/3 pyramidal neuron's response to a 10 ms laser pulse (blue) stimulation of POm axons during fentanyl sedation. (**B**) Population average of L2/3 response to POm, VPM, M1, and S2 input under fentanyl sedation (POm, n = 15 cells; VPM, n = 10 cells; M1, n = 8 cells; S2, n = 9 cells). Gray, average

*Figure 5 continued on next page*

*Figure 5 continued*

responses of individual neurons; Colored, population average. Baselines just prior to photo-activation have been aligned to allow visual comparison of responses. (C) Power-spectrum of persistent period (colored) vs. spontaneous period (black) in whole-cell recordings. Shading, 95% confidence bands computed using the Jackknife.

DOI: https://doi.org/10.7554/eLife.44158.009

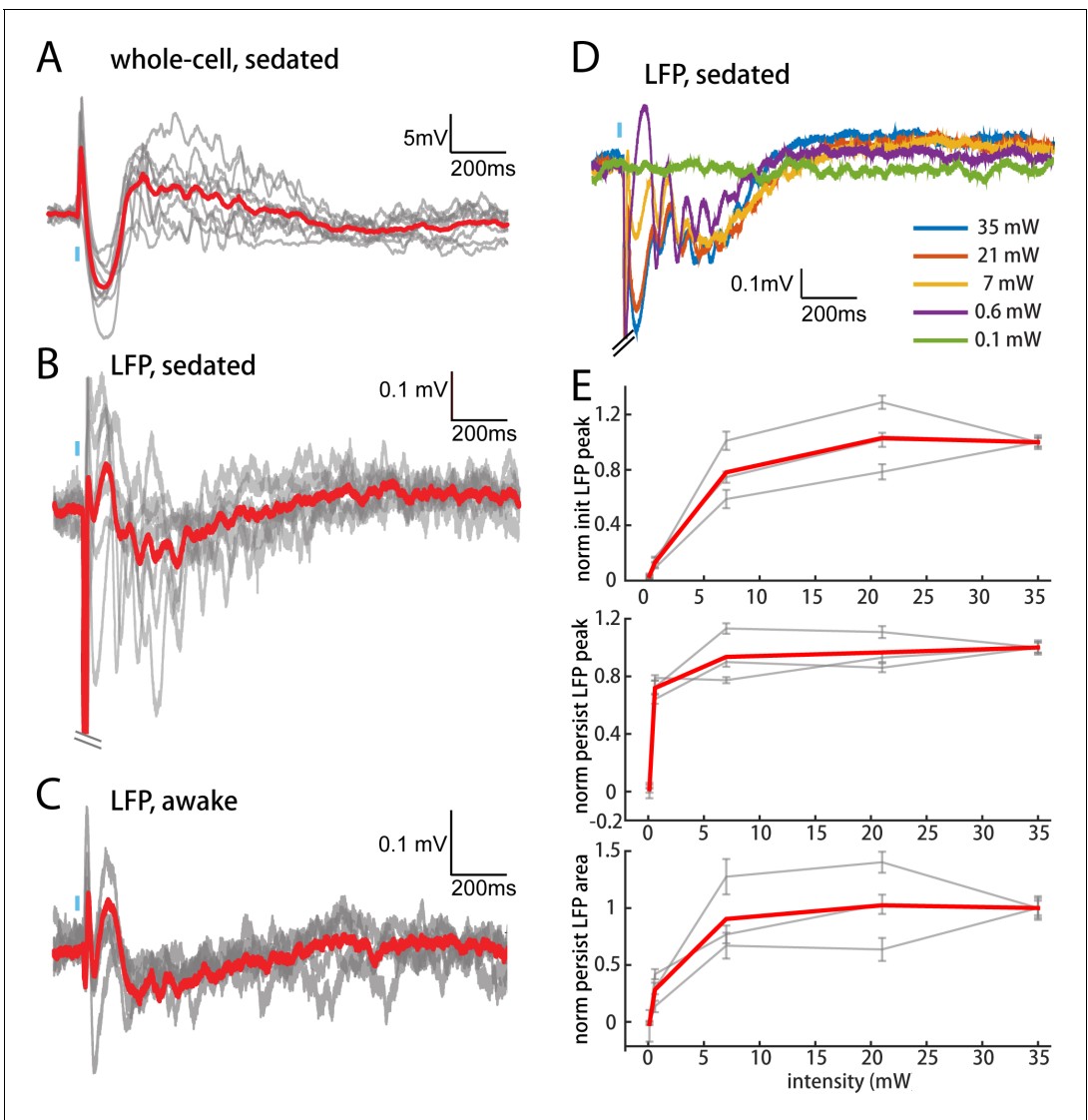

**Figure 6.** POm elicits an all-or-none persistent depolarization in L2/3 during wakefulness. (A) Population average of L2/3 whole-cell response to POm during fentanyl sedation (n = 15 cells). Gray, average responses of individual neurons. Red, population average. Baselines just prior to photo-activation have been aligned to allow visual comparison of responses. (B) L2/3 LFP response to POm inputs recorded in fentanyl sedated rats (n = 7 rats, 40 trials/rat). Gray, average responses at each recording site. Red, population average. (C) L2/3 LFP response to POm inputs recorded in awake rats (n = 4 rats). Gray, average responses at each recording site. Red, population average. (D) Example L2/3 LFP responses to POm stimulation with different levels of light intensity, recorded under fentanyl sedation. (E) Top: LFP initial peak amplitude vs. light intensity, normalized to responses induced by highest light intensity. Middle: LFP persistent peak amplitude vs. light intensity, normalized to responses induced by highest light intensity. Bottom: LFP persistent area vs. light intensity, normalized to responses induced by highest light intensity. Gray, average response for each rat. Red, population average (n = 3 rats; errorbar, SEM).

DOI: https://doi.org/10.7554/eLife.44158.010

## Persistent depolarization of L2/3 requires thalamic circuitry

POm may not only trigger but also sustain the persistent depolarization. We tested this by juxtasomally recording from individual POm neurons while photo-activating their thalamocortical axons in barrel cortex (*Figure 7A*). Because the persistent depolarization was seen in L2/3 in sedated but not anesthetized animals, we first compared POm spiking responses under sedation alone and with the volatile general anesthetic isoflurane (*Figure 7B*). We have previously used a similar strategy in order to rapidly alter the preparation and study individual cells under both conditions (*Constantinople and Bruno, 2011*). Spontaneous discharges of the POm neurons decreased more than ten-fold when anesthesia was induced (sedated mean ±SEM, 13.2 ± 8.7 Hz; anesthetized mean ±SEM, 1 ± 1.3 Hz; paired t-test, p=8 × $10^{-4}$; *Figure 7C*). The activity observed here during isoflurane-induced anesthesia is similar to that for urethane-anesthetized rats (*Masri et al., 2008*). Antidromic POm spiking responses to photostimulation were slightly but not significantly higher under sedation (paired t-test, p=0.09; *Figure 7D*). Given that POm synapses facilitate (*Viaene et al., 2011b*), suppressed POm spiking during anesthesia readily explains the smaller fast-onset PSPs compared to sedation (*Figure 4A*).

Consistent with POm generating the persistent depolarization in L2/3, five out of nine POm neurons exhibited persistent spiking responses to brief light activation of their terminals that was statistically (p<0.01) greater than spontaneous activity (*Figure 7E*). Nevertheless, long-lasting depolarization of L2/3 could conceivably be generated without somatic spiking of POm, such as by sustained axonal discharges (*Sheffield et al., 2011*) or recurrent circuits within cortex. To verify that

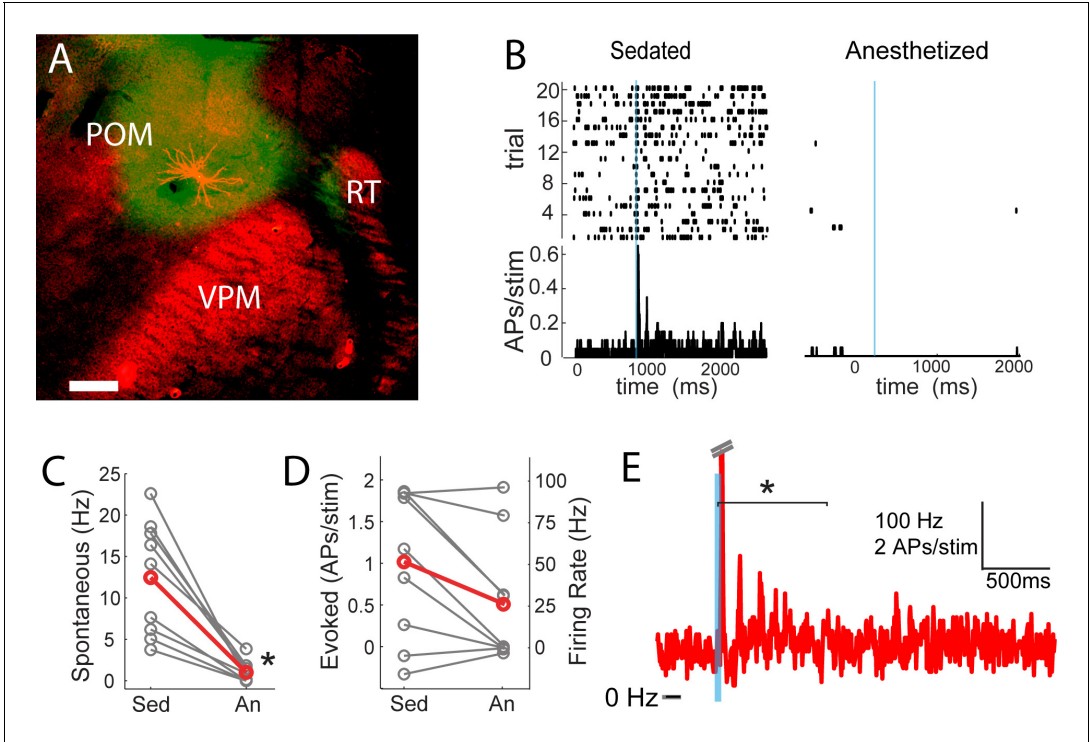

**Figure 7.** POm spiking mirrors L2/3 depolarization dynamics during anesthesia and sedation. (**A**) Morphologically recovered POm neuron recorded and filled juxtasomally in vivo. Red, biocytin-Alexa594. Green, ChR2-YFP. (**B**) Raster plot (top) and PSTH (bottom) of a POm neuron's response to photo-stimulation of ChR2-containing cortically-projecting axons. Left, POm cell recorded under sedation; Right, same cell under isoflurane-induced anesthesia. Blue line, 10 ms laser stimulation. (**C**) POm spontaneous activity is significantly lower under anesthesia than under sedation (paired t-test, n = 9, p=8×$10^{-4}$). Gray, individual cells; Red, mean. (**D**) Photo-activation of POm cortically-projecting fibers elicits more antidromic spikes in POm under sedation than anesthesia (paired t-test, n = 9, p=0.091). Peak firing rate is baseline-corrected by subtracting spontaneous firing rate measured in the pre-laser period. Gray, individual cells; Red, mean. (**E**) Population PSTH of POm neurons that displayed persistent activation under sedation (n = 5). Each cell displays significant higher firing rate than baseline (t-test, p<0.01) during the persistent period (0–800 ms post light stimulation, indicated by the bracket). Dashes, peak truncation for clarity.

DOI: https://doi.org/10.7554/eLife.44158.011

delayed persistent depolarizations in L2/3 are truly POm-dependent, we recorded L2/3 LFP responses to light while silencing POm somata with 10% lidocaine or 1 mg/ml muscimol injected in POm (*Figure 8A*). We found that the initial response in L2/3 remained intact and may have even grown during inactivation (*Figure 8B,C*), perhaps due to relief from POm-driven cortical inhibition. The delayed persistent depolarization, however, was abolished upon inactivation (*Figure 8B,D*), ruling out potential artifact of channelrhodopsin activation of POm-S1 axon terminals. Moreover, this result demonstrates that the delayed persistent depolarization in L2/3 is generated through recurrent circuitry involving POm, rather than a synaptic mechanism or a purely intracortical mechanism.

## Discussion

L2/3 pyramidal neurons in the rodent barrel cortex are sparsely active under a large range of conditions, including when awake animals perform simple tactile tasks. This suggests that bottom-up

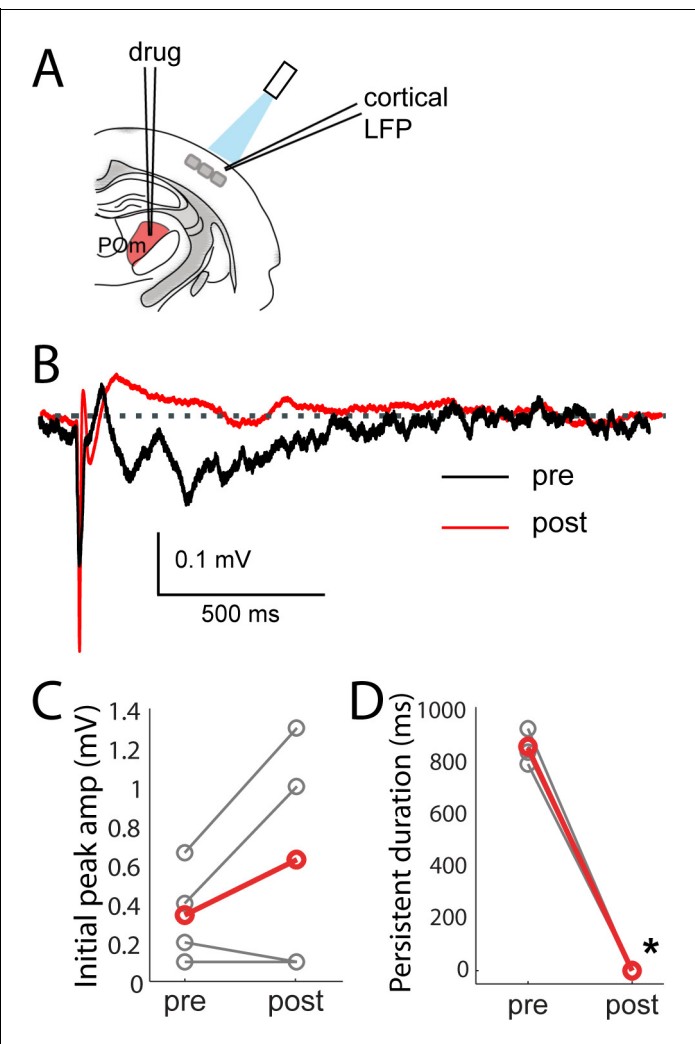

**Figure 8.** Persistent depolarization of L2/3 requires sustained POm firing. (**A**) Schematic of POm inactivation experiment. POm fibers are photo-activated while cortical LFP is recorded, before and after injection of 10% lidocaine or 1 mg/mL muscimol through a pipette in POm. (**B**) Population average of L2/3 LFP responses to POm input pre (black) and post (red) inactivation (n = 4 rats, 40 trials/rat). (**C**) Pharmacological inactivation slightly increased the magnitude of initial response in L2/3 (paired t-test, n = 4 rats, 40 trials/rat, p=0.24). Gray, individual animals; red, population average. (**D**) Persistent depolarization in L2/3 was abolished by POm inactivation (paired t-test, p<$10^{-4}$). Gray, individual animals; red, population average.
DOI: https://doi.org/10.7554/eLife.44158.012

sensory information from the periphery is insufficient to strongly drive L2/3 neurons, possibly resulting in a sparse code for tactile information. Alternatively, additional inputs such as long-range inputs from various higher order cortical and subcortical regions may be needed to unleash L2/3 activity. POm, M1 and S2 have axons that ramify in L1 of barrel cortex and overlap with apical dendrites of L2/3 pyramidal neurons. All three areas could potentially ungate L2/3 activity. In spite of this, M1 and S2 inputs were relatively weak, having little impact on L2/3 pyramids. Only POm inputs appeared sufficient to drive L2/3 neurons and boost their sensory responses.

Additionally, our study provides a direct anatomical comparison of these three long-range projections. The overall laminar distributions of POm, M1, and S2 projections to barrel cortex here are consistent with previous studies of the individual projections (*Kinnischtzke et al., 2014*; *Minamisawa et al., 2018*; *Ohno et al., 2012*; *Veinante and Deschênes, 2003*; *Wimmer et al., 2010*). We found that these three areas differ in their infragranular targets. POm, S2 and M1 axons together cover the infragranular layers, from their most shallow depth in L5A to their deepest in L6B. Of note, the S2 zone appears to correspond to the region of L5/6 that lacks POm and VPM innervation (*Wimmer et al., 2010*). These results suggest that within L5 and 6, there potentially exist multiple sub-layers that receive and integrate information from different cortical and subcortical regions.

For both our functional and anatomical results, an important question is the extent to which we sufficiently infected source regions. The whisker representations in POm and S2 are similarly small (~0.5 mm wide) and have known locations, lateral of VPM and S1, respectively (*Carvell and Simons, 1987*; *Fabri and Burton, 1991a*; *Fabri and Burton, 1991b*; *Kwegyir-Afful and Keller, 2004*). Our single-track multi-bolus methods of infecting POm and S2 readily achieved ChR2 expression over this spatial extent. Additionally, POm and S2 axons arborize broadly in S1 across many columns and are similarly loosely topographic (*Minamisawa et al., 2018*; *Ohno et al., 2012*). If only a very specific location within S2 were capable of input similar in magnitude to POm, one would have expected large S2 inputs in at least a subset of our experiments, but such S2 magnitudes were never observed. Neither geometry nor connectivity appears likely to account for our POm and S2 results.

M1 is a large structure, but again the whisker-associated areas of M1 are smaller in extent. Pilot experiments initially exploited physiological mapping techniques to optimize M1 infection (see Materials and methods), but these produced the same small M1 inputs as stereotaxic injections. We therefore blanketed M1 with multiple multi-bolus tracks during stereotaxic virus delivery. Whereas we had to avoid contaminating VPM and S1 in the case of POm and S2, we could afford to infect larger frontal areas, potentially beyond M1, to ensure that virtually all of M1 was expressing ChR2. Despite the larger volumes of our M1 infections, POm inputs exhibited the larger amplitudes.

Further studies are nonetheless needed to explore select thalamic sub-regions or cortical cell types and to test whether they differ in influence on sensory processing. POm may contain multiple anterior and posterior subnuclei that preferentially innervate L5 versus L1 of barrel cortex, respectively (*Ohno et al., 2012*). Similarly, M1 cortico-callosal cells send axons which ramify in both L1 and L5/6 of barrel cortex, whereas corticofugal cells send more collaterals to L1 of barrel cortex (*Veinante and Deschênes, 2003*). It is unknown if these subregions and cell types are always active at the same time. Our anatomical and physiological data likely mixed the two pathways, both the L1- and L5/6-preferring projections. Additionally, L2/3 has been suggested to contain separate subcircuits that are active at behaviorally distinct times (*Chen et al., 2013*), and such subcircuits may be more strongly or weakly targeted by POm. Indeed, POm is known to provide greater synaptic input to the most active L2/3 cells (*Jouhanneau et al., 2014*).

Consistent with the stronger POm effect we observed, a previous study of primary visual cortex in anesthetized primates found that pharmacological activation of pulvinar, the visual analog of POm, could enhance L2/3 neuronal responses to visual stimuli (*Purushothaman et al., 2012*). Also in line with the weak M1 impact here, M1 is known to strongly activate inhibitory neurons in L2/3 of barrel cortex (*Kinnischtzke et al., 2014*; *Lee et al., 2013*; *Zagha et al., 2013*). Locomotion enhances L2/3 responses in mouse V1 and suppresses primary auditory cortex, but these effects may involve neuromodulation of cortical inhibitory cells as opposed to direct M1 excitatory synapses on L2/3 pyramidal neurons (*Fu et al., 2014*; *Zhou et al., 2014*). There has been limited investigation of S2 inputs to barrel cortex. We found that S2 provides only minor depolarizing inputs to L2/3 pyramidal neurons, perhaps slightly stronger than the contribution of M1. A recent study suggests that S2

feedback had little or no effect on overall discharges of L2/3 units in S1 and may be more useful for modulating directional tuning of S1 neurons (*Minamisawa et al., 2018*).

Long-range inputs have been suspected of enhancing sensory processing via dendritic nonlinearity. Coincident activation of apical and basal dendrite inputs to L5 neurons can engage large calcium spiking events in vivo, which results in supralinear integration (*Xu et al., 2012*). We observed only linear and sublinear summation of potential top-down inputs (POm, M1, S2) with bottom-up sensory signals (whisker stimuli) in L2/3 pyramidal neurons, suggesting the absence of voltage-gated calcium spikes in their dendrites. This conclusion is consistent with recent imaging studies of L2/3 cells in S1 and V1 that found sensory stimulation produces sparse local dendritic 'hot spots' but not global events (*Palmer et al., 2014*). Linear summation also indicates that L4 sensory inputs and POm inputs likely innervate separate dendritic compartments of L2/3 cells, whereby local depolarization by one group of synapses minimally affects the local driving force and input resistance of another dendritic compartment. This type of summation is consistent with L4 axons mainly innervating L2/3 basal dendrites over apical tufts (*Lübke et al., 2003*) and POm preferentially synapsing on L1 apical tufts over basals (*Petreanu et al., 2009*).

Several possible mechanisms could explain how POm exerts stronger influence than S2 and M1 over L2/3 pyramidal cells even though they all innervate L1 to roughly the same degree. First, POm synapses onto L2/3 cells could simply be stronger. However, slice studies of the individual pathways using optogenetic stimulation have not revealed a major difference between M1- and POm-evoked currents (*Audette et al., 2018*; *Kinnischtzke et al., 2014*), though these pathways have yet to be directly compared in vitro. Second, differences in their subcellular locations could conceivably lead to selective engagement of the nonlinear mechanisms mentioned above. High-throughput light microscopy approaches would be useful for scanning whole apical tufts to assess possible differences (*Chandrasekaran et al., 2015*; *Schoonover et al., 2014*). Third, POm selectively targets 5HT3AR inhibitory cells and in vitro evokes particularly delayed inhibition in L2 relative to excitation (*Audette et al., 2018*). The delayed inhibition affords POm an unusually long window of integration in cortex. M1 and S2 might target other inhibitory cell types, endowing them with more typical, short integration windows.

## Persistent depolarization

POm activation could persistently depolarize cortical L2/3 pyramidal neurons for up to about 1 s. One possible explanation is that our photo-activation protocol induces a barrage of action potentials generated locally in the POm axons in cortex rather than by the axon initial segment near the POm somata. Axonal barrage firing has been observed in hippocampal and cortical interneurons (*Sheffield et al., 2011*; *Suzuki et al., 2014*). Another possibility is that persistent depolarization is generated by recurrent circuitry within L2/3 or between L2/3 and L5. Our pharmacological inactivation of POm indicated that activation of POm somata is necessary for generating persistent depolarization in the cortex. This rules out both axonal barrage firing and purely intracortical circuit mechanisms.

Since POm cells have little or no recurrent connectivity with one another (*Deschênes et al., 1998*; *Ohno et al., 2012*), the persistent firing of POm is likely due to a loop involving POm and one or more other structures. We observed a characteristic alpha/beta-range oscillation in the persistent depolarization, which is reminiscent of oscillations such as sleep spindles, which are generated through interactions between primary thalamic relay cells and the reticular inhibitory neurons (*von Krosigk et al., 1993*). The persistent depolarization we observed may be produced by similarly reciprocally connected structures, such as POm and possibly cortical layer 5/6 and/or the thalamic reticular nucleus, perhaps with some contribution of zona incerta. Indeed POm input can prolong suprathreshold whisker response of L5 cells on a similar timescale in isoflurane-anesthetized mice (*Mease et al., 2016*). The POm input to the cortex might then act directly through projections on excitatory cells as well as indirectly through inhibitory neurons, which contribute to visual cortex oscillations overlapping the beta frequency range (*Veit et al., 2017*). Recent modeling work demonstrates that brief bursts of apical tuft input combined with broad proximal dendrite input may contribute to beta oscillation generation (*Sherman et al., 2016*), and POm synapses are anatomically positioned to exploit precisely such a mechanism. POm activity increases during active movement (*Moore et al., 2015*; *Urbain et al., 2015*), which may be a key condition for inducing these long-lasting depolarizations in L2/3 during behavior.

The functional consequences of the persistent depolarization are potentially important for both plasticity and perception. POm might not only permit L2/3 response to sensory stimulation, but also sensitize L2/3 for prolonged periods thereafter. First, POm was recently shown to facilitate an NMDA-dependent plateau potential that can induce synaptic long-term potentiation in layer 2/3 without somatic spikes for a period on the order of ~100 ms following the stimulus (*Gambino et al., 2014*). Second, whisker contact alone in an awake mouse does not persistently depolarize L2/3, but, with training on a simple tactile detection task, a prolonged secondary depolarization of unknown origin emerges. This late depolarization correlates with behavioral choices and a small increase of L2/3 firing rates (albeit still sparse), and optogenetic inhibition of barrel cortex during this time period interferes with performance (*Sachidhanandam et al., 2013*). The time courses of the late depolarization during behavior and our POm axon stimulation results are similar, suggesting a possible POm substrate. This also raises the question of whether POm activation of cortical neurons is crucial for perception of sensory stimuli.

There are multiple possibilities for how excitation might flow through these circuits. Suppressing barrel cortex activity abolishes the responses of POm neurons to whisker stimulation (*Diamond et al., 1992*). This might have suggested a simple latching scenario, in which barrel cortex activates POm via L5 descending axons and then POm persistent firing provides feedback that sustains cortical activity. However, we found that persistent depolarizations of L2/3 via POm were absent during anesthesia. Additionally, our results and others have shown that overall POm activity is highly elevated during wakefulness (*Masri et al., 2008*; *Urbain et al., 2015*), possibly due to cholinergic regulation of zona incerta inputs to POm (*Masri et al., 2006*). An alternative possibility then is that persistent POm input to L2/3 has multiple triggers. In addition to synapses from trigeminal brainstem and S1, POm receives connections from motor cortex and posterior parietal cortex in multiple species (*Neylon and Haight, 1983*). During active behavior, any of these inputs might conceivably trigger persistent firing in the now more excitable POm circuit, which then prolongs a window of enhanced cortical responsiveness even when there was no preceding sensory stimulus. This could support multiple behaviors.

Secondary thalamic nuclei, such as POm and pulvinar, have been postulated to be important intermediaries of communication between cortical areas (*Sherman and Guillery, 2011*), subserving sensory-motor associations and other high-order aspects of sensory behavior. Lesion studies in non-human primates as well as humans have demonstrated that pulvinar is important for selective attention in visual search (*Bender and Butter, 1987*; *Ungerleider and Christensen, 1979*; *Ward et al., 2002*; *Wilke et al., 2010*). Behavioral studies further demonstrated that responses of pulvinar neurons are stronger when stimuli are presented within an attended region of visual space (*Petersen et al., 1985*). Pulvinar neurons corresponding to the attended location also fire tonically at elevated levels prior to stimulus presentation, suggesting that pulvinar may be involved in allocating spatial attention and/or maintaining working memory (*Saalmann et al., 2012*). This elevated activity might reflect the engagement of an intrathalamic or thalamocortical loop as we evoked in POm by channelrhodopsin stimulation. Intriguingly, activity in pulvinar and visual cortical areas is highly coherent in beta-range frequencies (*Saalmann et al., 2012*) similar to those we observed (*Figure 5*). Our results raise the possibility that beta oscillations may be a general cortical signature of interactions with secondary thalamus.

Few analogous behavioral studies have been done to elucidate the functional role of POm. One behavioral study concluded that POm lesions do not affect an animal's ability to detect passive deflection of a single whisker (*Narumi et al., 2007*). However, POm may be primarily engaged during active sensing with multiple whiskers given its large receptive fields and tight reciprocal connections with M1 and S1 (*Diamond et al., 1992*; *Groh et al., 2014*; *Groh et al., 2008*; *Miyashita et al., 1994*; *Trageser and Keller, 2004*). Physiological studies have demonstrated that POm sensory responses can be significantly boosted by the presence of neuromodulators (*Masri et al., 2006*), thus raising the possibility that POm activity could be strongly modulated by attention. Additionally, subsets of POm neurons are highly responsive to noxious stimuli (*Masri et al., 2009*). However, rather than selectively representing pain, POm may respond to any stimuli of high behavioral salience. Secondary sensory nuclei in thalamus may enable encoding of such high-order contextual information during behavior (*Saalmann et al., 2012*) via the long-lasting input they provide to L2/3. This may render L2/3 sensitive to conjunctions of sensory and contextual inputs and provide an eligibility trace enabling plasticity over behavioral timescales.

# Materials and methods

**Key resources table**

| Reagent type (species) or resource | Designation | Source or reference | Identifiers | Additional information |
| --- | --- | --- | --- | --- |
| Antibody | rabbit anti-GFP primary | EMD Millipore | AB3080P | |
| Antibody | goat anti-rabbit Alexa 488 secondary | Invitrogen | A11034 | |
| Recombinant DNA reagent | AAV1.hSyn.ChR2(H134R)-eYFP.WPRE.hGH | U.Penn Vector Core | | |
| Chemical compound, drug | lidocaine HCl, monohydrate | Sigma Aldrich | L5647 | |
| Chemical compound, drug | muscimol | Sigma Aldrich | M1523 | |
| Chemical compound, drug | strepavidin-Alexa594 | Life Technologies | S-11227 | |

All procedures were approved by the Institutional Animal Care and Use Committee at Columbia University (protocols AC-AAAD0960 and AC-AAAF9003).

## Optogenetics

To photo-activate long-range projection fibers, female ~70–100 g Wistar rats (Charles River) were injected with adeno-associated virus (AAV1) to express a ChR2-eYFP fusion protein driven by the human synapsin promoter (AAV1.hSyn.ChR2(H134R)-eYFP.WPRE.hGH, U.Penn Vector Core). Note that effects or effect sizes could conceivably differ for male rats. Standard aseptic technique was used. Rats were anesthetized with isoflurane (1–3% in $O_2$) and placed in a stereotax using blunt earbars. Pre-emptive systemic analgesia was administered (carprofen, 5 mg/kg, subcutaneous). Ophthalmic ointment was applied to the eyes. Rectal body temperature was maintained at 37°C with a heating pad. A small incision was made in the scalp to expose the skull overlying the target structure. The skull was thinned with a dental drill.

Craniotomies were placed stereotaxically: M1, 1.5–2.5 mm anterior to bregma and 1–2.5 mm lateral from midline (injections 500–1500 µm beneath the pia); S2, 2–3 mm posterior and 6–6.5 mm lateral (1500–2000 µm); and POm, 2.5–3.5 mm posterior and 2.5–4.0 mm lateral (4500–5000 µm). VPM was targeted by physiologically locating its C and D whisker row representations and injecting at approximately this location. POm was targeted by injecting 700 µm medial of those VPM representations.

Injections were made to achieve similar coverage of each of the four structures, which have different geometry. For M1, four different injection tracts spaced 500 µm apart in a square pattern were used. For each tract, a 40–60 nl volume was ejected at depths of 1500, 1000 and 500 µm from the pia, for a grand total of 12 injections. Only one injection tract was made through S2, POm, and VPM, which are smaller than M1. POm received only two 40 nl ejections at two depths 300 µm apart along a single injection tract, as did VPM. S2 received three injections at depths of 1500, 1000 and 500 µm along its single injection tract made at the stereotaxic site.

Virus was injected over ~20 min using a Nanoject II auto-nanoliter injector (Drummond, Broomall, PA). The craniotomies were covered with bone wax, and the incision closed with absorbable sutures. Animals were allowed to recover from surgery in a clean cage with softened food palettes and water overnight before returning to their home cage.

Because M1 is larger than POm and S2, we attempted to refine M1 injection locations based on mapping procedures. In a small number of pilot experiments, we placed M1 injections by first 1) recording from M1 with electrodes to identify regions responsive to the center of the whisker field, typically whisker C2, 2) microstimulating M1 to evoke movements of whiskers in the center of the whisker pad, such as C2, and 3) flavoprotein imaging of M1 during electrical stimulation of L1 or L2 above the C2 barrel in S1 (*Shibuki et al., 2003*). Each of these physiological mapping techniques indicated circumscribed subregions of M1 that were already within our stereotaxic windows and, when targeted for infection, subsequently yielded M1 inputs to L2/3 that were <2 mV in peak amplitude. Consequently, we used only stereotaxic location for placing subsequent M1 infections.

After ~3 weeks of viral expression, animals were prepared for physiology experiments as described below. Light-activation of infected fibers was achieved by placing a 200 μm fiber optic immediately above a craniotomy over barrel cortex and delivering isolated 10 ms pulses of 473 nm light using a DPSS laser (OEM) controlled by a mechanical shutter. Fiber output (~38 mW) was checked between experiments using a power meter. We measured the average light response of a neuron with 20–40 trials of light stimulation with 2 s interstimulus intervals. For combined light and whisker stimuli, onset of whisker stimulus precedes the light stimulus by 7 ms (approximate synaptic delay from periphery to L2/3), to maximize the coincidence of whisker- and light-evoked potentials arriving in L2/3.

## Animal preparation for physiology

In all cases, animals were initially anesthetized with isoflurane (1–3% in $O_2$). Body temperature was kept at 37°C by a heating blanket. Eyes were coated with lubricating ointment to prevent drying. The parietal and occipital bones were exposed, and a metal post for positioning the head was attached to the skull using dental acrylic. The parietal bone overlying left barrel cortex (centered 2.5 mm posterior to bregma and 5.5 mm lateral of the midline) was thinned with a dental drill until transparent, and small craniotomies (<0.5 mm$^2$) were made over the thinned region. The dura was removed.

For urethane anesthesia experiments, 74 female Wistar rats (150–250 g) were used. After the above surgery was complete, animals were administered urethane by IP injection (0.9–1 g/kg). For sedation experiments, 12 rats were additionally prepared as described previously (*Bruno and Sakmann, 2006*). Briefly, cannulae were inserted into the trachea (for mechanical ventilation), femoral artery (for blood pressure monitoring) and jugular vein (for drug infusion). Screws were inserted in the right frontal and parietal bones for electrocorticogram ('EEG') recording. All wounds were infiltrated with bupivacaine. Fentanyl (~10 μg/kg/hr) and pancuronium bromide (1.6 mg/kg/hr) were continuously infused after discontinuation of general anesthesia, and rats were ventilated (90–100 breaths/min). Mean arterial blood pressure was typically ~120 mm Hg.

Four animals were prepared for the anesthetized-awake preparation as described previously (*Constantinople and Bruno, 2011*). Prior to making craniotomies, screws were inserted in the right frontal and parietal bones for electrocorticogram ('EEG') recording. Instead of switching to urethane or fentanyl, animals remained on isoflurane anesthesia. Rats were wrapped in a blanket and secured in a plastic tube to reduce movement. The local anesthetic bupivacaine (0.5%) was regularly applied to the area of the head surrounding the acrylic. To avoid startling the rat, a black curtain was placed around the air table, and noise in the lab minimized.

## Electrophysiology

Patch pipettes (4–7 MΩ) were pulled from borosilicate glass and tip-filled with (in mM) 135 K-gluconate, 10 HEPES, 10 phosphocreatine-Na$_2$, 4 KCl, 4 ATP-Mg, 0.3 GTP, and 0.2–0.4% biocytin (pH 7.2, osmolarity 291). Pipette capacitance was neutralized prior to break-in, and access resistance was 10–60 MΩ. Recordings were digitized at 32 kHz. Similar pipettes were used for juxtasomal recording of POm neurons. Juxtasomal pipettes were filled with 4% biocytin in aCSF (in mM: 135 NaCl, 5.4 KCl, 1.8 CaCl$_2$, 1.0 MgCl$_2$, and 5.0 HEPES; pH 7.2). After acquiring single-unit data, we attempted to fill the recorded neurons by injecting square current pulses (1–3 nA, 250 ms on, 250 ms off) for several minutes. LFP pipettes (2–3 MΩ) were filled with aCSF. LFPs were bandpassed 1–325 Hz.

Pharmacological inactivation of POm: 90–100 nL of muscimol (1 mg/mL) or Lidocaine (10%) was injected over ~20 min using a Nanoject II auto-nanoliter injector (Drummond, Broomall, PA). LFP recordings were obtained ~30 min post injection.

## Whisker stimulation

Individual whiskers were deflected using multi-directional piezoelectric stimulators. Whiskers were positioned inside the stimulator ~10 mm from the base of the hair and deflected 5.7° (1 mm amplitude) using relatively high-velocity (onset and offset: ~570°/sec) ramp-and-hold movements. Deflections were applied randomly in each of eight directions, in 45° increments relative to the horizontal alignment of the rows. A receptive field was mapped by applying 10–20 blocks of such stimuli (80–160 total stimuli with 2 s interstimulus intervals).

## Histology

After recordings, rats were deeply anesthetized and perfused transcardially with cold 0.1 M sodium phosphate buffer (PB) followed by 4% paraformaldehyde. The left barrel cortex was cut tangentially in 100 μm sections to the white matter. The rest of the left hemisphere was sliced coronally in 100 μm sections. Tangential sections were stained with streptavidin conjugated to Alexa 594 (Life Technologies) to visualize recorded neurons. In tangential and coronal sections, ChR2-eYFP signal was amplified by using a rabbit anti-GFP primary antibody (at 1:1000, incubated overnight in 5% normal goat serum and 1% Triton-X in PB at 4°C) and a goat anti-rabbit-Alexa488 secondary antibody (at 1:200, for 2 hr in 5% normal goat serum and 1% Triton-X in PB at room temperature, Invitrogen). Using epifluorescence or confocal microscopy, we confirmed the presence of infected somata in the targeted brain region and infected axons near recorded S1 neurons.

The location of a L2/3 cell relative to its barrel center was measured by 3D reconstruction in Neurolucida (MicroBrightfield). The radial trunk axons of the recorded L2/3 neurons were visibly well filled in the L4 sections. Its location was marked, and the borders of the L4 barrel it passed through were traced. The horizontal distance between the axon and the centroid of the barrel borders were then measured in the same section.

## Analysis

Required n was estimated prior to the study by standard power analysis assuming an effect size (difference in PSP amplitudes) of 2 mV, SD of 1.5 mV, and 80% power, suggesting a minimum n of approximately 10 neurons per group. No outliers were excluded.

Data were analyzed using custom Matlab routines. Initial and persistent depolarizations to photostimulation were measured from the mean trace. Persistent depolarization was classified as present/absent by testing whether the maximum over 150–1000 msec following the laser pulse onset was statistically significantly different from a 100-msec period preceding the laser pulse. The onset of the persistent response is defined to be the time at which the trace crosses the baseline value (prior to the laser pulse) in a positive direction after the IPSP. The offset is defined to be the time at which the trace returned to baseline.

Parametric tests were used for data that appeared Normally distributed. Nonparametric tests were used otherwise. Spectral analysis was performed with Chronux. DC was subtracted from $V_m$ prior to spectral analysis.

## Acknowledgements

We thank Michael Shadlen, Chris Rodgers, Amanda Kinnischtzke, Y Kate Hong, Akash Khanna, and Georgia Pierce for comments on the manuscript. We thank Drew Baughman for technical support. This project was supported by NINDS RO1 NS069679, NINDS R01 NS094659, the Klingenstein Fund for Neuroscience, and the Rita Allen Foundation (RMB).

## Additional information

### Funding

| Funder | Grant reference number | Author |
| --- | --- | --- |
| National Institute of Neurological Disorders and Stroke | RO1 NS069679 | Randy M Bruno |
| Rita Allen Foundation | | Randy M Bruno |
| Esther A. and Joseph Klingenstein Fund | | Randy M Bruno |
| National Institute of Neurological Disorders and Stroke | R01 NS094659 | Randy M Bruno |

The funders had no role in study design, data collection and interpretation, or the decision to submit the work for publication.

## Author contributions
Wanying Zhang, Conceptualization, Formal analysis, Investigation, Writing—original draft, Writing—review and editing; Randy M Bruno, Conceptualization, Formal analysis, Supervision, Funding acquisition, Writing—original draft, Writing—review and editing

## Author ORCIDs
Randy M Bruno https://orcid.org/0000-0002-5122-4632

## Ethics
Animal experimentation: This study was performed in strict accordance with the recommendations in the Guide for the Care and Use of Laboratory Animals of the National Institutes of Healths. Experiments were performed according to protocols approved by the Institutional Animal Care and Use Committee of Columbia University.(protocols AC-AAAD0960 and AC-AAAF9003).

## Decision letter and Author response
Decision letter https://doi.org/10.7554/eLife.44158.017
Author response https://doi.org/10.7554/eLife.44158.018

# Additional files

## Supplementary files
• Transparent reporting form
DOI: https://doi.org/10.7554/eLife.44158.013

## Data availability
There are no reagents that could be deposited. All physiological and anatomical data will be placed on a public website and will also be available by direct request.

The following dataset was generated:

| Author(s) | Year | Dataset title | Dataset URL | Database and Identifier |
|---|---|---|---|---|
| Wanying Zhang, Randy M Bruno | 2019 | Data from: High-order thalamic inputs to primary somatosensory cortex are stronger and longer lasting than cortical inputs | https://dx.doi.org/10.5061/dryad.hj4pb70 | Dryad Digital Repository, 10.5061/dryad.hj4pb70 |

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
