## [Decision Letter]

[Editors’ note: a previous version of this study was rejected after peer review, but the authors submitted for reconsideration. The first decision letter after peer review is shown below.]

Thank you for submitting your work entitled "Higher-order thalamus, but not cortex, strongly activates primary somatosensory cortex" for consideration by *eLife*. Your article has been reviewed by three peer reviewers, and the evaluation has been overseen by a Reviewing Editor (BRE) and a Senior Editor. The following individuals involved in review of your submission have agreed to reveal their identity: Mathew E Diamond (Reviewer #3).

Our decision has been reached after consultation between the reviewers. While at least two reviewers and the BRE appreciate the importance of your approach and the potential novelty of your results, e.g., "… POm's effect is largely excitatory despite projections to inhibitory L1 is surprising from a purely anatomical point of view and is in agreement with recent functional demonstration of the excitatory role of the POm to S1 pathway.", the comments of reviewer 1 and particularly reviewer 2 raise strong concerns that prevent us from considering your manuscript for publication at this time. The key issues are:

(Reviewer 2) Location of PO thalamus: The input to "Barrel" cortex from PO thalamus lies at the approximate ventral edge of PO thalamus, which abuts VPM thalamus. Thus it is a matter of necessity to show all injections and carefully verify the injection sites. We note that Cyt-ox staining, which highlights VPM over PO thalamus, is an effective means of demonstrating the border.

(Reviewer 1 and Reviewer 2) The need to use VPM input as a baseline level: All inputs are noted as weak in the manuscript. This should be relative to the dominant input to the whisker column, which comes from VPM thalamus. These data would also provide strong contact with a larger body of past literature.

(Reviewer 1) Extreme hyperpolarization of cells in cortical layers 2/3: As noted by reviewer 1, these intracellular voltage levels appear aberrant and "could have the effect of making the cortical neurons hyperexcitable and therefore over-represent the sensory gating phenomenon". This needs to be checked and explained.

(Reviewer 2) The extent of activation of CHR2 as a function of depth needs to be addressed. In this regard, the authors may consider the use of C1V1, ReaChR, or Chrimson to improve the depth dependence of cortical activations of inputs for L2/3 cells.

We appreciate that considerable time and effort may well be involved in forming experiments to address these issues. While rejecting your paper in its current form, the editors would be open to reconsidering a thoroughly augmented manuscript as a new submission given the overall interest in the topic and potential for the results.

Reviewer #1:

Zhang and Bruno show that the higher-order whisker thalamus POm provides potent excitatory input to layer 2/3 neurons in the primary sensory barrel cortex, such that POm input coincident with sensory (whisker stimulation) input evokes spiking in layer 2/3. The functional "gating" of L2/3-a cortical layer so often studied but so difficult to make spike-by POm is an exciting and profound finding, and is an important step towards understanding thalamocortical interactions during sensory processing. The report that POm's effect is largely excitatory despite projections to inhibitory L1 is surprising from a purely anatomical point of view and is in agreement with recent functional demonstration of the excitatory role of the POm to S1 pathway. Furthermore, the presented results contrast the strong POm effect to the weaker effect of cortical inputs from S2 and also from motor cortex. Zhang and Bruno further strengthen POm's role in providing excitation on long timescales and they built on these findings by demonstrating that POm-specific effects are seen in sedated and awake but not anesthetized animals, which the authors suggest as a possible mechanism for cognitive functions, e.g. memory and learning.

In addition to addressing the very timely issue of the role of higher-order thalamus on cortical dynamics, the experiments are of high quality and are well designed and controlled. The study combines single-cell electrophysiology, anatomical techniques, and optogenetic and pharmacological manipulation of the circuits in question in anesthetized and sedated rats, as well as LFP measurements in awake rats.

The paper is clearly and concisely written and for the most part, data and analysis are well presented and summarized. However, I have a few questions regarding basic aspects of the data and analysis, which make some findings difficult to interpret, along with some other points which would strengthen and clarify the paper.

1) The membrane potentials shown in Figure 2 seem quite hyperpolarized – well below -80 mV for each case in the example raw data, and in the summary plot, approximately half of the recordings have baseline potentials of less than -75 mV and approximately one quarter have baseline potentials well below -80 mV. Perhaps I missed it, but I could not find comments in the manuscript on these low RMPs, which seem surprising especially in vivo, even when assuming strong and tonic inhibition of these circuits.

• My concern is that this level of hyperpolarization could have the effect of making the cortical neurons hyperexcitable and therefore over-represent the sensory gating phenomenon, if it is due to some effect of the preparation which also affects juxtasomal spike recordings and not only whole-cell recordings. Did the authors obtain any preliminary data on POm's effects on neurons in other layers for comparison?

• If any whole-cell spiking data showing the combination of whisker stimulation and POm input was collected, was there a correlation between the magnitude of sensory boost and baseline membrane potential? If possible, it would also be useful to plot the reversal potential of POm inputs from single to investigate mixed inhibition and excitation.

2) The stimuli – both whisker and optogenetic Pom – are not particularly physiological. Especially given the lab's previous use of dynamic, more complicated stimuli, it would be useful to indicate how one can extrapolate from the present results to more realistic patterns of input. This issue seems particularly relevant for the POm input and long timescale effect, as the authors state that they attempted to introduce ChR into as much of the nucleus as possible; therefore the light stimulus will presumably induce massive synchronized spiking in POm. It is an interesting point that the long timescale effect seems to be all or none, in contrast to the more graded initial effect, but it is hard to interpret what this widespread activation of POm (and POm's additional inputs e.g. in motor cortex and S2) would mean for more precise patterns of activity vs. the network being driving into an extreme of oscillatory behavior.

3) Based on anatomical overlap alone, the three regions could equally cause "gating," as mentioned in the Discussion. However, the functional findings point to a situation where "Peter's rule" does not apply. The authors do discuss some reasons for the discrepancy, but this discussion should be expanded upon especially in light of functional predictions from anatomical data (bouton/dendrite overlap) in the literature.

4) As Figure 1A shows that the axonal density in L2/3 varies considerably, it is surprising that there was no significant correlation between EPSP amplitudes and the recorded neurons' locations with respect to these septal patches. Did the authors check the relative amount of YFP fluorescence surrounding the recorded neurons)? If EPSP amplitude doesn't covary with expression in L2/3, then doesn't this suggest that the input goes into the tuft?

Reviewer #2:

In this study, the authors use mainly optogenetic techniques to test the relative strength of three inputs to cells of the rat's barrel cortex. The inputs are from M1, S2, and POm. They conclude that the POm input is strong and the others, weak. Although the methodology is fairly straightforward, there are just too many flaws, of varying seriousness, to recommend publication. Poor experimental design allows many alternative explanations for the results, many of which are uninteresting. While each problem alone might be fixable, the sheer number is daunting.

1) Interpretation of the data requires that the ChR2 be placed accurately. This is especially important for the POm activation. POm abuts VPM, and so any spillover could mean that VPM inputs were also activated, which would completely invalidate the results. The documentation for this (Figure 1A lower) is completely inadequate. The injections sites need to be shown for every case and the border between POm and VPM must be defended and illustrated.

2) The photoactivation is applied to the surface of cortex. How deep does it penetrate? Are only layer 1 terminals hit? According to Figure 1B, the main peaks are 1mm down in barrels. Are terminals there stimulated? However, Figure 1A shows that POm dominates septa in upper layers (a Figure 1B version of septa would be useful), and these POm fibers would be selectively activated by the photostimulation. Thus an uncontrolled variable would activate many more septal terminals from POm, and only if such activation does not affect the recordings in barrels would this not be a problem. This seems a dubious point at best.

3) The authors repeatedly refer to POm input as "strong", "potent" and: "Only POm inputs appeared sufficient to drive L2/3 neurons…" Actually, the data do not support very strong input here. It seems more appropriate to describe this input as the least weak among the three tested. One would expect the VPM input to be strong, and if they wish to make such claims, they should compare the POm to VPM inputs (see also below).

4) In the third paragraph of the subsection “POm facilitates sensory responses of L2/3 pyramidal neurons”, the authors describe the location of recorded cells with respect to barrel centers. How was this determined?

5) Figure 7B makes no sense to me, which may be my misreading, but I do find it worrisome. If I understand correctly, photostimulation of the axons in cortex leads to an antidromic spike that is seen in the left histogram, but why is none seen in the right, even if background firing is reduced. Why should anesthesia affect antidromic activation? Also, if one antidromically activates thalamocortical cells as I assume is done here, one should see a monosynaptic early peak (as in the left histogram but not the right) followed by a disynaptic dip due to activation of TRN cells that would inhibit the recorded cell. This dip is missing in the left histogram. Why?

6) The authors state, "POm was targeted by physiologically locating the C and D whisker row representations…," presumably maximizing the connectivity with barrel cortex. No mention is made of similar alignment procedures for the M1 or S2 inputs, raising yet another uncontrolled variable that could explain the apparent greater strength of the POm input.

7) Figure 5 curiously includes VPM stimulation. This is odd and troubling to say the least. No mention of methodology here and no reference to relative VPM strength as suggested in point #3 above.

Reviewer #3:

On this interesting and solid manuscript I would not recommend any additional experiments but would suggest some optimization of the presentation.

There is a sort of conflicting narrative behind the results. The story begins by saying, point blank, L2/3 cells fire sparsely:

Abstract – "Layer (L) 2/3 pyramidal neurons in the primary somatosensory cortex (S1) are sparsely active, spontaneously and during sensory stimulation."

Introduction – "Layer (L) 2/3 pyramidal neurons in the primary sensory cortices exhibit sparse activity, both spontaneously and in response to sensory stimuli (Barth and Poulet, 2012). Even in awake animals performing tactile detection tasks, L2/3 firing probability remains low and is substantially lower than that of most other cortical layers (O'Connor et al., 2010)."

And even the outset of Discussion – "L2/3 pyramidal neurons in the rodent barrel cortex are sparsely active under a large range of conditions, including when awake animals perform simple tactile tasks."

But then in the Discussion, in the third paragraph of the subsection “Persistent Depolarization”, all the emphasis is on the significance of persistent activity in superficial layers which seems discordant with the preceding sections. The phrasing in various sections could be reworked a bit to make the Discussion more harmonious with the earlier sections, or vice versa.

In sum, the narrative is:

1) – Abstract, Introduction. Layer 2/3 cells fire sparsely.

2) Results. POm stimulation can depolarize the cells and cause them to fire (Figure 4D and E in particular).

3) Discussion. Well, we already knew that in certain behaviors there is widespread long-lasting depolarization in L2/3, which disagrees with 1.

The authors also ought to discuss the ramifications of POm's possible dependence on descending projections from barrel cortex for its own excitability (as indicated in much older literature, at least in the anesthetized brain). If that circuit – which by the way should not be lumped together with Sherman's view of POm in cortex-to-cortex communication – operates in the awake animal, it means that POm would require a descending input from layer 5 to get significantly excited (perhaps due to convergence with slower ascending trigeminal inputs) and that POm's firing would then allow subsequent excitation of cortical layer 2/3. Whether the above summary is accurate, or whether a more likely scenario can be depicted, the authors should address this issue of the temporal flows of excitation suggested by their work.

[Editors’ note: what now follows is the decision letter after the authors submitted for further consideration.]

Thank you for submitting your article "Unique properties of high-order thalamic inputs versus cortical inputs to primary somatosensory cortex" for consideration by *eLife*. Your article has been reviewed by three peer reviewers, and the evaluation has been overseen by Ronald Calabrese as the Senior and Reviewing Editor. The reviewers have opted to remain anonymous.

The reviewers have discussed the reviews with one another and the Reviewing Editor has drafted this decision to help you prepare a revised submission.

Summary:

The manuscript is substantially improved over the last version, with the major scientific and technical concerns resolved. The authors study how cortical and thalamic input can modify S1 L2/3 responsiveness and conclude that secondary thalamus (POm) can open a long-duration window of enhanced L2/3 responsiveness, while inputs from cortical M1 and S2 is relatively weak. From a conceptual point of view, this is important work, since we still are just at beginning stages to understand thalamo-cortical circuitry.

There are still a few issues that need to be addressed by revision but they need not necessitate re-review.

Essential revisions:

1) The remaining experimental concern is whether a direct comparison between S2, M1, and POm is wholly warranted. We are satisfied that the authors can detect whether POm inputs are well-labeled, but are less convinced by their ability to target the anatomically correct region of S2 and M1 with saturating levels of ChR2 expression. S2 in particular is a broad region, and injections that are slightly off the barrel column analyzed (for PW stimulation) will result in weak light-evoked responses. This is experimentally tough – since every animal will have slightly different experimental coordinates for maximal S2 label in the selected PW), but must be introduced as a major caveat in the interpretation/Discussion of the data. This concern is also the case for M1.

2) Generally, the study is well-conceived and will be of great interest in the field. The first 4 and last 4 figures remain somewhat disconnected, and it would be very helpful to try to draw a more coherent link between the two parts of the study.

---

## [Author Response]

[Editors’ note: the author responses to the first round of peer review follow.]

Our decision has been reached after consultation between the reviewers. While at least two reviewers and the BRE appreciate the importance of your approach and the potential novelty of your results, e.g., "… POm's effect is largely excitatory despite projections to inhibitory L1 is surprising from a purely anatomical point of view and is in agreement with recent functional demonstration of the excitatory role of the POm to S1 pathway.", the comments of reviewer 1 and particularly reviewer 2 raise strong concerns that prevent us from considering your manuscript for publication at this time. The key issues are:(Reviewer 2) Location of PO thalamus: The input to "Barrel" cortex from PO thalamus lies at the approximate ventral edge of PO thalamus, which abuts VPM thalamus. Thus it is a matter of necessity to show all injections and carefully verify the injection sites. We note that Cyt-ox staining, which highlights VPM over PO thalamus, is an effective means of demonstrating the border.

This is a good point. Our study contains dozens of rats for any one area alone – a complete gallery even at postage stamp size is pages of figures. We have therefore provided a new supplementary figure that contains a gallery of example POm injections that illustrate for the reader 1) the general location we targeted, 2) the size of the POm infection, and 3) the size of VPM contamination, if any. More directly to R2’s point, this figure shows new quantitative analyses of the degree to which intended POm infection versus unintended VPM contamination might produce our results. This new Figure 4—figure supplement 1 and a new Results paragraph document that POm infection size, but not VPM contamination, correlates with the amplitude of synaptic inputs we measured. Thus, VPM contamination cannot account for our results. POm appears to be the main determinant of the large PSPs in layer 2/3.

(Reviewer 1 and Reviewer 2) The need to use VPM input as a baseline level: All inputs are noted as weak in the manuscript. This should be relative to the dominant input to the whisker column, which comes from VPM thalamus. These data would also provide strong contact with a larger body of past literature.

We have refined the text and title to avoid overstating the strength of POm input and frame the work as a relative comparison of different inputs. We also elaborate on the VPM data, providing VPM statistics and pointing out that VPM inputs are ~7 times stronger than POm inputs in terms of the initial peak.

(Reviewer 1) Extreme hyperpolarization of cells in cortical layers 2/3: As noted by reviewer 1, these intracellular voltage levels appear aberrant and "could have the effect of making the cortical neurons hyperexcitable and therefore over-represent the sensory gating phenomenon". This needs to be checked and explained.

The intracellular voltage levels are not unusual of L2/3 cells under these conditions and recorded with reasonably good access resistance for in vivo recording (here, 10-60 MOhm). Our sample of L2/3 whole-cell recordings in urethane-anesthetized rats had a median membrane potential of roughly -75 mV (Figure 2E), which is not extremely hyperpolarized for this cell type under these conditions. Brecht, Roth and Sakmann, 2003, reported a mean of -83 mV for L2/3 cells in urethane-anesthetized rats, and Petersen et al., 2003, reported a mean of -75 mV. The reviewer may be thinking of recent studies using awake, not anesthetized animals. With regards to that possibility, our own lab has shown that this cell type in awake rats is more depolarized, albeit only slightly (to the -70 to -65 mV range), because the protracted down states seen under anesthesia largely disappear during wakefulness (Constantinople and Bruno, 2011). The range (min and max) over which the neurons fluctuate is the same whether the animal is awake or anesthetized.

The slightly more depolarized level during wakefulness is precisely why we additionally added the fentanyl sedation experiments, in which the animals are not anesthetized and have membrane potential levels and dynamics as in awake animals.

Furthermore, biophysically, the exact level of depolarization would affect all 3 classes of inputs (POm, S2, and M1) equivalently and cannot explain the relative differences in magnitude.

(Reviewer 2) The extent of activation of CHR2 as a function of depth needs to be addressed. In this regard, the authors may consider the use of C1V1, ReaChR, or Chrimson to improve the depth dependence of cortical activations of inputs for L2/3 cells.

All POm axons reconstructed to date have branches in L1 (Deschenes et al., 1998; Ohno et al., 2012; Oberlaender, Ramirez, and Bruno, unpublished data). In the rat, 95% of the motor cortex neurons that project to primary somatosensory cortex have been observed to have branches in L1 (Veinante and Deschenes, 2003). Thus, strong illumination of L1 in S1 is expected to activate virtually all infected neurons projecting to S1. As shown in our study (Figure 7) and as shown by others, activating terminals triggers backpropagating action potentials, which will invade the rest of the axonal tree and soma. In other words, there are few if any additional neurons that can be recruited by increasing light penetration or repeating our entire study (measurements of three different groups of animals) with a red-shifted opsin.

Additionally, the original text related to Figure 4B explained that, if POm’s innervation of the septum explained its strong input to L2/3, one would expect to the size of the PSP to be a function of distance from the barrel center. As shown in Figure 4B, this is not the case.

Reviewer #1:[…] The paper is clearly and concisely written and for the most part, data and analysis are well presented and summarized. However, I have a few questions regarding basic aspects of the data and analysis, which make some findings difficult to interpret, along with some other points which would strengthen and clarify the paper.1) The membrane potentials shown in Figure 2 seem quite hyperpolarized – well below -80 mV for each case in the example raw data, and in the summary plot, approximately half of the recordings have baseline potentials of less than -75 mV and approximately one quarter have baseline potentials well below -80 mV. Perhaps I missed it, but I could not find comments in the manuscript on these low RMPs, which seem surprising especially in vivo, even when assuming strong and tonic inhibition of these circuits.

Because the editor felt this was a particularly important point, we have addressed it above in response to their summary. As we describe above in greater detail, these values are not unusual. We nevertheless now comment on them and their consistency with previous studies in the Results.

• My concern is that this level of hyperpolarization could have the effect of making the cortical neurons hyperexcitable and therefore over-represent the sensory gating phenomenon, if it is due to some effect of the preparation which also affects juxtasomal spike recordings and not only whole-cell recordings. Did the authors obtain any preliminary data on POm's effects on neurons in other layers for comparison?

We did not examine other layers, but other layers have different passive and active physiological properties (among other differences) and are not a neutral comparison. However, to the reviewer’s main concern, the original submission reported the input resistances of the L2/3 cells (Figure 2F), which were ~40 MOhm on average and typical of other studies. Our L2/3 recordings are not in an unusually hyperexcitable state. We now explicitly comment on this.

• If any whole-cell spiking data showing the combination of whisker stimulation and POm input was collected, was there a correlation between the magnitude of sensory boost and baseline membrane potential? If possible, it would also be useful to plot the reversal potential of POm inputs from single to investigate mixed inhibition and excitation.

Figure 4D, E showed analyses of spiking data for combined whisker and POm stimulation versus whisker stimulation alone. However, we detected no significant relationship of the magnitude of the boost in spiking (spiking with laser – spiking without laser) and the baseline membrane potential (linear regression, p = 0.15). This is potentially complicated by the stochasticity of spike generation, the sublinearity of synaptic integration (see below), opposing effects of baseline on driving force for excitation versus inhibition, and cell-to-cell differences in excitation versus inhibition. One likely requires a particularly large sample to detect such effects.

As requested, we regressed POm input magnitude against baseline membrane potential. This predicts a reversal potential for POm inputs of approximately -34 mV, suggestive of mixed inhibition and excitation and consistent with other studies of synaptic pathways studied in vivo. We have added this in the Results.

2) The stimuli – both whisker and optogenetic Pom – are not particularly physiological. Especially given the lab's previous use of dynamic, more complicated stimuli, it would be useful to indicate how one can extrapolate from the present results to more realistic patterns of input. This issue seems particularly relevant for the POm input and long timescale effect, as the authors state that they attempted to introduce ChR into as much of the nucleus as possible; therefore the light stimulus will presumably induce massive synchronized spiking in POm. It is an interesting point that the long timescale effect seems to be all or none, in contrast to the more graded initial effect, but it is hard to interpret what this widespread activation of POm (and POm's additional inputs e.g. in motor cortex and S2) would mean for more precise patterns of activity vs. the network being driving into an extreme of oscillatory behavior.

We are grateful to the reviewer for raising this interesting question, which we did not expound upon in our original submission. The dynamic, more complicated stimuli that the reviewer refers to is our paper Ramirez et al., 2014, which used an array of whisker stimulators to explore more naturalistic stimulus space in a controlled manner. An extrapolation is indeed possible. Ramirez et al. showed that inputs evoked by each whisker are 1) linearly integrated for small PSPs and sublinearly integrated for large PSPs, seemingly due to cellular driving force constraints (Ramirez Figure 7D), but 2) the effect of surround whiskers is still facilitatory, leading to a net boost over principal whisker stimulation alone (Ramirez Figure 7C). Interestingly, this is exactly the pattern of results we obtained for POm and principal whisker stimulation: sublinear integration that remains net facilitatory (this manuscript Figure 3B, C). In other words, POm input appears to be similar to a sensory input in terms of integration by and impact on L2/3 cells. Because the L2/3 cells do not spike to the various cycles of the POm input – the input alone is subthreshold – the network is not necessarily expected to enter a state of extreme oscillation when additional stimuli are introduced.

One would therefore expect that POm would, overall, enhance spiking responses to spatiotemporally complex patterns of whisker stimuli similar to our single-whisker stimuli used here. This is reminiscent of an attention-like enhancement thought to exist for visual pulvinar. Of course, our data do not rule out the possibility that POm could enhance specific spatiotemporally complex patterns and suppress others. This is an interesting hypothesis and worthy of a future study. We now comment on the relationship between these results and our previous study and discuss possible extrapolations.

3) Based on anatomical overlap alone, the three regions could equally cause "gating," as mentioned in the Discussion. However, the functional findings point to a situation where "Peter's rule" does not apply. The authors do discuss some reasons for the discrepancy, but this discussion should be expanded upon especially in light of functional predictions from anatomical data (bouton/dendrite overlap) in the literature.

Our original submission briefly suggested that POm was so effective for reasons demonstrated in the slice by Alison Barth’s lab (Audette et al., 2017). We have moved this sentence to the part of the Discussion just following our discussion of nonlinear mechanisms in apicals and written a more comprehensive discussion of why POm might have stronger effects than M1 or S2:

“Several possible mechanisms could explain how POm exerts stronger influence than S2 and M1 over L2/3 pyramidal cells even though they all innervate L1 to roughly the same degree. […] The delayed inhibition affords POm an unusually long window of integration in cortex. M1 and S2 might target other inhibitory cell types, endowing them with more typical, short integration windows.”

4) As Figure 1A shows that the axonal density in L2/3 varies considerably, it is surprising that there was no significant correlation between EPSP amplitudes and the recorded neurons' locations with respect to these septal patches. Did the authors check the relative amount of YFP fluorescence surrounding the recorded neurons)? If EPSP amplitude doesn't covary with expression in L2/3, then doesn't this suggest that the input goes into the tuft?

The lack of correlation with distance to the septum, while a result, is understandable: As we discussed in our response to the editor’s summary, all POm axons reconstructed to date branch extensively in L1 (Deschenes et al., 1998; Ohno et al., 2012; Oberlaender, Ramirez, and Bruno, unpublished data) *but* have little or no branching as they ascend the septum on their way to L1. Additionally, a single axon often has long segments that ascend the septal spaces between more than one pair of barrels. The epifluorescence is therefore misleading in this respect because many unbranched axons are being concentrated into small spaces, creating a strong signal that does not necessarily translate into a complex arbor with many synapses. We now comment on this point.

Furthermore, in our original manuscript we tried to make exactly the point that the POm synapses are mainly targeting the apical tuft, but we realize now that we were too subtle on this point. We have revised the Discussion to make this second point much more explicit, additionally including references to previous work examining the subcellular distributions of synaptic contacts:

“Linear summation also indicate that L4 sensory inputs and POm inputs likely innervate separate dendritic compartments of L2/3 cells, whereby local depolarization by one group of synapses minimally affects the local driving force and input resistance of another dendritic compartment. This type of summation is consistent with L4 axons mainly innervating L2/3 basal dendrites over apical tufts (Lubke et al., 2003) and POm preferentially synapsing on L1 apical tufts over basals (Petreanu et al., 2009).”

Reviewer #2:In this study, the authors use mainly optogenetic techniques to test the relative strength of three inputs to cells of the rat's barrel cortex. The inputs are from M1, S2, and POm. They conclude that the POm input is strong and the others, weak. Although the methodology is fairly straightforward, there are just too many flaws, of varying seriousness, to recommend publication. Poor experimental design allows many alternative explanations for the results, many of which are uninteresting. While each problem alone might be fixable, the sheer number is daunting.1) Interpretation of the data requires that the ChR2 be placed accurately. This is especially important for the POm activation. POm abuts VPm, and so any spillover could mean that VPm inputs were also activated, which would completely invalidate the results. The documentation for this (Figure 1A lower) is completely inadequate. The injections sites need to be shown for every case and the border between POm and VPm must be defended and illustrated.

As explained at the beginning of this point-by-point reply, we have provided a gallery of injections and new analyses that demonstrate that VPM contamination cannot account for our results.

2) The photoactivation is applied to the surface of cortex. How deep does it penetrate? Are only layer 1 terminals hit? According to Figure 1B, the main peaks are 1mm down in barrels. Are terminals there stimulated? However, Figure 1A shows that POm dominates septa in upper layers (a Figure 1B version of septa would be useful), and these POm fibers would be selectively activated by the photostimulation. Thus an uncontrolled variable would activate many more septal terminals from POm, and only if such activation does not affect the recordings in barrels would this not be a problem. This seems a dubious point at best.

We addressed this issue in detail in our overall reply. In short, virtually all POm and M1 axons arborize in layer 1 of S1 where the light is most intense. Therefore every infected axon is expected to be activated independent of how little or extensive it arborizes in deeper layers.

Additionally, light intensity at a depth of 1 mm in the brain is estimated to be only ~1% of the intensity at the surface (Aravanis et al., 2007). In other words, blue light is likely to be more restricted to L1 terminals than red light, which would increase not decrease variability if it were not the case that virtually all POm and M1 axons branch in L1.

3) The authors repeatedly refer to POm input as "strong", "potent" and: "Only POm inputs appeared sufficient to drive L2/3 neurons…" Actually, the data do not support very strong input here. It seems more appropriate to describe this input as the least weak among the three tested. One would expect the VPM input to be strong, and if they wish to make such claims, they should compare the POm to VPM inputs (see also below).

We now clarify throughout the text that we are making a relative comparison rather than an absolute statement of strength. We report additional VPM numbers as requested below.

4) In the third paragraph of the subsection “POm facilitates sensory responses of L2/3 pyramidal neurons”, the authors describe the location of recorded cells with respect to barrel centers. How was this determined?

We added the following to the Histology part of the Materials and methods section:

*“*The location of a L2/3 cell relative to its barrel center was measured by 3D reconstruction in Neurolucida (MicroBrightfield). The radial trunk axons of the recorded L2/3 neurons were visibly well filled in the L4 sections. Its location was marked, and the borders of the L4 barrel it passed through were traced. The horizontal distance between the axon and the centroid of the barrel borders were then measured in the same section.”

5) Figure 7B makes no sense to me, which may be my misreading, but I do find it worrisome. If I understand correctly, photostimulation of the axons in cortex leads to an antidromic spike that is seen in the left histogram, but why is none seen in the right, even if background firing is reduced. Why should anesthesia affect antidromic activation? Also, if one antidromically activates thalamocortical cells as I assume is done here, one should see a monosynaptic early peak (as in the left histogram but not the right) followed by a disynaptic dip due to activation of TRN cells that would inhibit the recorded cell. This dip is missing in the left histogram. Why?

The left and right panels illustrate the same cell under two different conditions. Antidromic spikes traveling along an axon are not guaranteed to fire a somatic spike, which would be necessary for us to detect an action potential by juxtasomal recording. Also, like most general anesthetics, isoflurane’s mechanism(s) of action are not fully understood. Isoflurane is known to bind numerous neurotransmitter receptors and likely also indirectly affects the neuromodulatory environment. The anesthesia-induced changes to the intrinsic properties, overall level of hyperpolarization, and possible modulation of active conductances are likely opposing the triggering of somatic spikes in the POm somata.

We agree that one would expect TRN-mediated inhibition in POm, though perhaps not necessarily to the extent seen in VPM, which may be what the reviewer is thinking of. One can see a clear period of silence lasting 100-200 ms in many individual trials of the raster plot. However, there are several trials with bursts around the time of the light, which means that the PSTH will lack the empty bins that the reviewer is expecting to see. That is, the average hides what can be seen in individual trials.

Moreover, we should underscore that the point of Figure 7 was to show whether or not POm cells were also firing persistently. This was not guaranteed as axonal barrage firing has been described for some types of axons (Sheffield et al., 2011). That is, the axon could be bursting without somatic bursts. We showed this is not the case for POm, which led us to the experiments to test the necessity of POm in the persistent firing phenomenon (Figure 8).

6) The authors state, "POm was targeted by physiologically locating the C and D whisker row representations…," presumably maximizing the connectivity with barrel cortex. No mention is made of similar alignment procedures for the M1 or S2 inputs, raising yet another uncontrolled variable that could explain the apparent greater strength of the POm input.

The physiological targeting procedure was used because POm has different geometry to M1 and S2 and we were concerned about contaminating VPM. M1 is a larger, sheet-like structure where one would also be less concerned about virus spreading beyond its borders. We made multiple injection tracts through M1, but only a single injection tracts through POm, S2, and VPM. Despite M1 receiving many more boluses of virus, we observed little effect of its inputs on

S1. These results are also consistent qualitatively similar experiments regarding M1->S1 and S2->S1 inputs from other labs. Variability in virus injection does not seem able to explain our consistent results.

Also, in a few pilot experiments, we had targeted injections to M1 by flavoprotein imaging of the cortex while antidromically stimulating their axons in S1. This produced activity spots in the flavoprotein imaging where we would subsequently inject virus. This yielded synaptic input measurements that were similar to the stereotaxic approach, which we subsequently used for the study.

We now describe more completely the injection procedures for POm, VPM, S2, and M1.

7) Figure 5 curiously includes VPM stimulation. This is odd and troubling to say the least. No mention of methodology here and no reference to relative VPM strength as suggested in point #3 above.

VPM was included in experiments investigating persistent depolarization because POm’s difference from S2 and M1 in inducing persistent depolarization might simply be because it produced strong initial input than S2 and M1. VPM produces an even stronger initial volley and could therefore control for this possibility. As reviewer 1 noted, this is a nice control. We now motivate our use of VPM more clearly. We also added more technical detail of VPM injections to the Materials and methods and compare the initial volleys for POm to VPM in the Results.

Reviewer #3:On this interesting and solid manuscript I would not recommend any additional experiments but would suggest some optimization of the presentation.There is a sort of conflicting narrative behind the results. The story begins by saying, point blank, L2/3 cells fire sparsely:Abstract – "Layer (L) 2/3 pyramidal neurons in the primary somatosensory cortex (S1) are sparsely active, spontaneously and during sensory stimulation."Introduction – "Layer (L) 2/3 pyramidal neurons in the primary sensory cortices exhibit sparse activity, both spontaneously and in response to sensory stimuli (Barth and Poulet, 2012). Even in awake animals performing tactile detection tasks, L2/3 firing probability remains low and is substantially lower than that of most other cortical layers (O'Connor et al., 2010)."And even the outset of Discussion – "L2/3 pyramidal neurons in the rodent barrel cortex are sparsely active under a large range of conditions, including when awake animals perform simple tactile tasks."But then in the Discussion, in the third paragraph of the subsection “Persistent Depolarization”, all the emphasis is on the significance of persistent activity in superficial layers which seems discordant with the preceding sections. The phrasing in various sections could be reworked a bit to make the Discussion more harmonious with the earlier sections, or vice versa.In sum, the narrative is:1) Abstract, Introduction. Layer 2/3 cells fire sparsely.2) Results. POm stimulation can depolarize the cells and cause them to fire (Figure 4D and E in particular).3) Discussion. Well, we already knew that in certain behaviors there is widespread long-lasting depolarization in L2/3, which disagrees with 1.

First, we see that our exposition inadvertently created some confusion between persistent depolarization (a window of enhancement during which a cell is more likely to respond to a subsequent input) and persistent activity (a window during which a cell IS firing more). We had used the terms “persistent response” and “persistent depolarization” interchangeably numerous times throughout the text. “Response” of course connotes suprathreshold firing, which is not what we meant to convey. We now only use the term “persistent depolarization” with regards to L2/3. We also clarify that this is a window of enhanced opportunity for L2/3 responses, not that L2/3 is necessarily firing throughout this time.

Second, our Discussion paragraph referred to some previous results regarding late subthreshold depolarizations. Those *subthreshold* results do not disagree with claims of sparse *suprathreshold* discharges. Moreover, some of the subthreshold results we cite are on a shorter timescale from those we present here. We now clarify these points to prevent unintended contradictions of our earlier statements:

“The functional consequences of the persistent depolarization are potentially important for both plasticity and perception. POm might not only permit L2/3 response to sensory stimulation, but also sensitize L2/3 for prolonged periods of time thereafter. First, POm was recently shown to facilitate an NMDA-dependent plateau potential that can induce synaptic long-term potentiation in layer 2/3 without somatic spikes for a period on the order of ~100 ms following the stimulus (Gambino et al., 2014). Second, whisker contact alone in an awake mouse does not persistently depolarize L2/3, but, with training on a simple tactile detection task, a prolonged secondary depolarization of unknown origin emerges. This late depolarization correlates with behavioral choices and a small increase of L2/3 firing rates (albeit still sparse), and optogenetic inhibition of barrel cortex during this time period interferes with performance (Sachidhanandam et al., 2013). The time courses of the late depolarization during behavior and our POm axon stimulation results are similar, suggesting a possible POm substrate. This raises the question of whether POm activation of cortical neurons is crucial for perception of sensory stimuli.”

The authors also ought to discuss the ramifications of POm's possible dependence on descending projections from barrel cortex for its own excitability (as indicated in much older literature, at least in the anesthetized brain). If that circuit – which by the way should not be lumped together with Sherman's view of POm in cortex-to-cortex communication – operates in the awake animal, it means that POm would require a descending input from layer 5 to get significantly excited (perhaps due to convergence with slower ascending trigeminal inputs) and that POm's firing would then allow subsequent excitation of cortical layer 2/3. Whether the above summary is accurate, or whether a more likely scenario can be depicted, the authors should address this issue of the temporal flows of excitation suggested by their work.

We have added a new paragraph to the Discussion, which also serves as an excellent bridge between the discussion of the phenomena and possible circuitry and the discussion of the possible behavioral functions:

“There are multiple possibilities for how excitation might flow through these circuits. […] This could support multiple behaviors.”

[Editors' note: the author responses to the re-review follow.]

Essential revisions:1) The remaining experimental concern is whether a direct comparison between S2, M1, and POm is wholly warranted. We are satisfied that the authors can detect whether POm inputs are well-labeled, but are less convinced by their ability to target the anatomically correct region of S2 and M1 with saturating levels of ChR2 expression. S2 in particular is a broad region, and injections that are slightly off the barrel column analyzed (for PW stimulation) will result in weak light-evoked responses. This is experimentally tough – since every animal will have slightly different experimental coordinates for maximal S2 label in the selected PW), but must be introduced as a major caveat in the interpretation/Discussion of the data. This concern is also the case for M1.

The whisker representation in S2 is actually not a broad region: Whisker S2 measures only ~0.5 mm wide and lies directly lateral to the whisker representation in S1 (Carvell and Simons, 1987; Fabri and Burton, 1991; Kwegyir-Afful and Keller, 2004). S2 is therefore roughly the same size as POm. A size difference leading to limited S2 coverage cannot explain the apparent input difference and bolsters our case for a comparison. Furthermore, a recent study in the mouse shows that, while S2->S1 connections have a topographical organization, the organization is very loose (Minamisawa et al., 2018): any small (<100 micron wide) location in S2 projects to an S1 region on the order of a magnitude larger. Thus, our larger (hundreds of micron wide) viral infections of S2 would provide even more expansive coverage of S1.

Similarly, M1 is a broad region, but the area that provides S1 feedback is more limited in size. Because we were unconcerned with contaminating surrounding structures in the case of M1 (unlike POm and S2), we could blanket M1 with multiple viral infection tracks. We did try flavoprotein imaging, microstimulation, and electrophysiological recording in attempts to optimize the placement of M1 injections, but these targeted infections never resulted in POm-like input strengths in M1 experiments. We therefore used the simpler approach of making multiple viral infection tracks to ensure that, even if a topographically special location within M1 existed, we would not miss it.

To address this issue, we have added 2 paragraphs to the Discussion addressing these points. We also mention in the Materials and methods that we tried various techniques for identifying potentially optimal M1 infection sites but they never yielded such large inputs as we observed for M1, prompting us to proceed with stereotaxic M1 experiments.

2) Generally, the study is well-conceived and will be of great interest in the field. The first 4 and last 4 figures remain somewhat disconnected, and it would be very helpful to try to draw a more coherent link between the two parts of the study.

The conceptual points of the figures are as follows. Figure 1-3: POm inputs appear stronger than S2 and M1 inputs under anesthesia. Figure 4: This is true under both anesthesia and sedation, a very different state that well approximates wakefulness. Figure 5: Under this condition, POm but no other areas can trigger a persistent depolarization. Figure 6: The same persistent depolarization exists during wakefulness and is all-or-none. Figure 7: The difference is likely due to the massive effect of state on POm dynamics. Figure 8: POm sustained firing during the sedated and awake conditions are required for the long-lasting depolarization. Thus, POm input differs from cortical inputs in terms of its strength, time course, and oscillatory signature.

We find the progression here logical. Nevertheless, to address the reviewers’ concern, we have retitled the paper (see also reviewer 3’s first comment). We have also tweaked the Results text transitioning from one figure to the next as well as the figure titles to amplify this narrative.